# Convex Dataset Valuation for Post-Training

**Siqi Zeng** [1]  **Christopher Jung** [2]  **Rui Li** [2]  **Zhe Kang** [2]  **Ming Li** [2]  **Nima Noorshams** [2]  **Zhigang Wang** [2]
**Fuchun Peng** [2]  **Han Zhao** [1]  **Xue Feng** [2]

## Abstract

Improving LLM performance on downstream tasks sometimes requires leveraging auxiliary datasets during post-training. In practice, however, developers face constraints on compute, labeling, and licensing costs that preclude using all available data, necessitating principled dataset-level selection. These constraints are increasingly shaped by dataset marketplaces, where data acquisition is governed by budgets and negotiation. We study dataset valuation as a subset selection problem during LLM post-training. Our goal is to identify and weight auxiliary datasets so as to maximize target task performance given constrained budgets. We first show that commonly used gradient alignment scores provide a reasonable yet incomplete valuation signal, as they ignore redundancy among datasets. To address this, we propose a scalable convex dataset-level valuation method based on kernel mean matching (KMM) in gradient space, which jointly accounts for alignment with the target task and redundancy across auxiliary datasets. Through extensive experiments across diverse post-training settings and tasks, we show that our approach consistently outperforms existing valuation baselines, achieving stronger performance with low computational overhead. Our results position dataset valuation as a practical decision tool for post-training data selection in market-constrained large language model settings. The code is available at https://github.com/uiuctml/conve x_data_valuation.

*The dataset, model access, and experiments are managed by UIUC.* [1]Department of Computer Science, University of Illinois Urbana-Chamapign, Urbana, IL, USA [2]Meta, Menlo Park, CA, USA. Correspondence to: Siqi Zeng <siqi6@illinois.edu>.

*Proceedings of the 43rd International Conference on Machine Learning*, Seoul, South Korea. PMLR 306, 2026. Copyright 2026 by the author(s).

## 1. Introduction

As large language models (LLMs) scale, training data has become an explicit economic resource. High-quality datasets are increasingly curated, licensed, and traded, giving rise to emerging dataset marketplaces in which data owners monetize access and model developers acquire data under limited budgets and asymmetric information (Chen et al., 2025; Zhang et al., 2023). In this regime, training data is no longer freely available at scale: acquisition decisions are constrained by cost, licensing, and negotiation, rather than by storage or availability alone.

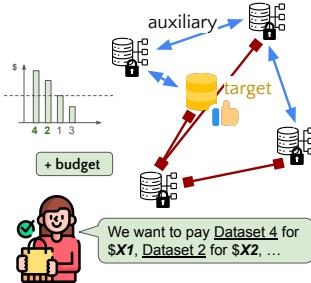

*Figure 1.* Buyers in data marketplace model. Auxiliary datasets exhibit heterogeneous positive and negative transfer relationships with respect to the target task (yellow) and with each other, reflecting redundancy and interference. Valuation scores provide a compact summary of these interactions, helping buyers rank candidate datasets and allocate budgets under limited data access.

In such dataset markets, model developers typically begin with a specific target task whose performance they aim to improve. But for many target tasks, obtaining sufficient high-quality supervision is challenging: labels may be scarce, expensive, or slow to collect, for example, when they require expert human annotation (Alzubaidi et al., 2023; McGiff & Nikolov, 2025). Consequently, practitioners often turn to auxiliary datasets as a practical means of enhancing target-task performance. From a buyer's perspective, this necessitates deliberate planning: developers must make discrete binary acquisition decisions, whether to include or exclude each auxiliary dataset, given limited resources and incomplete knowledge of their downstream impact.

In data marketplaces for large-scale model training, a central object is a dataset valuation signal: a summary of how

each candidate dataset is expected to contribute to target-task performance when combined with other available data. Such valuation is challenging because datasets interact non-trivially: auxiliary datasets may exhibit positive or negative transfer with respect to the target task and with each other (Rosenstein et al., 2005; Standley et al., 2020), so their value cannot be assessed in isolation.

As illustrated in Fig. 1, valuation signals enable buyer-side budgeting and acquisition planning. Recent work shows that valuation-aware pricing and acquisition strategies can substantially improve cost-performance trade-offs and the sustainability of data marketplaces, compared to heuristic approaches (Chen et al., 2025; Zhang et al., 2025). Despite this progress, existing dataset valuation methods are poorly aligned with the needs of post-training data markets. Many operate at the level of individual examples (Xia et al., 2024; Hu et al., 2025a;b; Dai et al., 2025; Wang et al., 2025; Ghorbani & Zou, 2019; Ilyas et al., 2022) or focus on pretraining data (Liu et al., 2025; Xie et al., 2023), making them ill-suited to post-training scenarios in which training objectives vary across tasks and acquisition decisions are made at the level of entire datasets. In such settings, naively aggregating sample-wise valuation scores to support dataset-level decisions is known to be unreliable (Hu et al., 2024).

**Contributions.** In this work, we propose a data valuation algorithm that provides a target-task-aware ranking that directly supports budgeting, acquisition planning, and tentative pricing in realistic post-training data marketplaces. We make the following contributions. (1) We formulate dataset valuation as a *post-training subset selection* problem for LLMs, motivated by budgeted data acquisition and dataset marketplaces, where each dataset represents a costly inclusion decision. (2) We propose a scalable and principled dataset-level valuation method based on gradient-space kernel mean matching (KMM), which accounts for both alignment with the target task and redundancy among multiple sources of auxiliary datasets, overcoming limitations of independent gradient-based scoring. The formulation leads to a convex quadratic program that can be efficiently solved with respect to the number of datasets. (3) We provide a finite-sample guarantee for the proposed method, under mild assumptions on the data, showing that the valuation scores can be reliably estimated from a finite number of samples from each dataset. (4) Through extensive experiments across diverse post-training settings, we demonstrate that our approach yields faithful and interpretable dataset rankings that consistently outperform existing valuation baselines across target tasks and hyperparameter choices.

## 2. Problem Setup

We consider the post-training setting of generative LLMs. Let $\theta_0 \in \mathbb{R}^d$ denote a pretrained model obtained from large-

scale pretraining. Our goal is to further adapt $\theta_0$ to a specific downstream task via post-training on a mixture of datasets. Let $D^{\text{tar}}$ denote a *target dataset* which defines the downstream task of interest yet assumed to be limited and difficult to obtain. Meanwhile, we are given a collection of *auxiliary datasets* $\mathcal{D}^{\text{aux}} = \{D_1, D_2, \ldots, D_N\}$, which may provide additional supervision during post-training. These datasets may differ in domain, language, style, or source, and their relevance to the target task is generally unknown.

**Dataset valuation** is the process of assigning to each auxiliary dataset $D_i$ a scalar score $w_i \in \mathbb{R}$, which reflects its expected contribution to post-training performance on a fixed target task, relative to other candidates in $\mathcal{D}^{\text{aux}}$. Dataset values should satisfy the following properties:

- **Task-dependent.** Values are defined with respect to a specific target task and evaluation metric. A dataset may be valuable for one task metric while being irrelevant or harmful for another.

- **Signed.** Values may be positive, zero, or negative, reflecting datasets that improve, have no effect on, or degrade target-task performance.

- **Comparative.** Values are meaningful only relative to other auxiliary datasets within the same post-training setting and are best interpreted as inducing a ranking over candidate datasets. Absolute prices are not assumed to transfer across tasks, models, or fine-tuning procedures.

### 2.1. Modeling Assumptions

**Post-training with auxiliary datasets.** We study dataset valuation in the post-training stage of LLMs, where a base model $\theta_0$ is adapted to a target task using supervised fine-tuning or reinforcement learning. In this setting, practitioners may augment limited target-task data $D^{\text{tar}}$ with auxiliary datasets $\mathcal{D}^{\text{aux}}$ to improve downstream performance. Given a fixed fine-tuning procedure $\text{FT}(\cdot)$, training on a selected set of auxiliary datasets $\mathcal{S} \subseteq \mathcal{D}^{\text{aux}}$ yields a post-trained model

$$\theta_{\mathcal{S}} = \text{FT}\big(\theta_0, \ \{D^{\text{tar}}\} \cup \mathcal{S}\big),$$

which is evaluated on the target dataset $D^{\text{tar}}$. Dataset valuation aims to guide the choice of $\mathcal{S}$ so as to improve target-task performance relative to the baseline model $\theta_{\text{tar}} = \text{FT}(\theta_0, \{D^{\text{tar}}\})$ trained on the target data alone.

**Dataset selection under budget constraints.** We focus on a binary selection problem: for each auxiliary dataset $D_i \in \mathcal{D}^{\text{aux}}$, the model developer must decide whether to include it in post-training or exclude it under budget constraints. This naturally induces a subset selection problem, where a decision is represented by the selected auxiliary subset $\mathcal{S} \subseteq \mathcal{D}^{\text{aux}}$. This assumption reflects common data marketplace practices (Chen et al., 2025), where datasets are acquired as indivisible units under one-time, upfront

licensing agreements. For any subset $\mathcal{S}$, let $u(\mathcal{S})$ denote the utility the buyer associates with selecting $\mathcal{S}$. Ideally, following the definition, the buyer would solve

$$\max_{\mathcal{S} \subseteq \mathcal{D}^{\text{aux}}, \, |\mathcal{S}| \leq k} u(\mathcal{S}). \tag{1}$$

However, the true $u(\mathcal{S})$ is typically unknown before acquisition, and evaluating it for exponentially many subsets is computationally intractable. Thus, in practice, a valuation method outputs dataset-level scores $w = (w_1, \ldots, w_N) \in \mathbb{R}^N$, and the buyer can make a tractable budgeted decision by selecting the top-$k$ datasets:

$$\mathcal{S}_w = \text{TopK}_{i \in [N]}(w_i). \tag{2}$$

The goal of dataset valuation is to produce scores whose induced selection $\mathcal{S}_w$ achieves high utility $\tilde{u}(\mathcal{S}_w)$.

**Limited access to auxiliary data prior to acquisition.** In practice, access to auxiliary datasets is constrained by legal and privacy considerations. As noted by Chen et al. (2025), data acquirers are often provided only with limited sample previews of a dataset before committing to its acquisition. Thus, during valuation, the full contents of an auxiliary dataset $D_i$ are not observable. Instead, the model developer has access only to a small size $m$ preview $\tilde{D}_i \subset D_i$, with the complete dataset becoming available only after the inclusion decision is made. Dataset valuation methods must therefore operate under partial observability of auxiliary data.

## 3. Methodology

### 3.1. Gradient-Based Approximation of Dataset Value

Let $\ell(x; \theta)$ denote the per-example loss of a model with parameters $\theta$ on input $x$. We define the population target and auxiliary risks as

$$\mathcal{L}_{\text{tar}}(\theta) = \mathbb{E}_{x \sim \mathcal{P}^{\text{tar}}}[\ell(x; \theta)], \qquad \mathcal{L}_i(\theta) = \mathbb{E}_{x \sim \mathcal{P}_i}[\ell(x; \theta)].$$

We first derive a simple gradient-based baseline under an independence assumption, where each dataset contributes only through its singleton utility:

$$u(\mathcal{S}) = \sum_{i \in \mathcal{S}} u(\{i\}). \tag{3}$$

It remains to specify the singleton utilities $u(\{i\})$. Consider fine-tuning a pretrained model $\theta_0$ with learning rate $\eta$. We define the singleton utility as the improvement in target loss after one gradient step on dataset $i$:

$$u(\{i\}) \triangleq \mathcal{L}_{\text{tar}}(\theta_0) - \mathcal{L}_{\text{tar}}(\theta_0 - \eta \nabla \mathcal{L}_i(\theta_0)). \tag{4}$$

A first-order Taylor approximation yields

$$u(\{i\}) = \eta \langle g_{\text{tar}}, g_i \rangle + O(\eta^2 \|g_i\|_2^2), \tag{5}$$

where $g_{\text{tar}} = \nabla \mathcal{L}_{\text{tar}}(\theta_0)$ and $g_i = \nabla \mathcal{L}_i(\theta_0)$.

Since $\eta > 0$ is a global constant, it does not affect the ranking of datasets. This yields the valuation rule

$$w_i = \langle g_{\text{tar}}, g_i \rangle. \tag{6}$$

More generally, instead of a single-step gradient, one may use trajectory-based *task vectors* (Ilharco et al., 2023), defined as the parameter difference $\tau_{\mathcal{S}} = \theta_{\mathcal{S}} - \theta_0$ resulting from fine-tuning $\theta_0$ on a dataset or dataset collection $\mathcal{S}$ for multiple steps. Task vectors capture longer-horizon training dynamics beyond the local linearization, and similarity between auxiliary and target tasks can be quantified via the dot product $\langle \tau_{\mathcal{S}}, \tau_{\text{tar}} \rangle$.

In the context of LLM post-training, dataset gradients are pooled at the sequence level before aggregation. For supervised fine-tuning (SFT), the per-example loss typically corresponds to sequence-level cross-entropy, computed as the average negative log-likelihood over generated tokens. For reinforcement learning (RL), $\ell$ denotes a surrogate loss derived from trajectory-level rewards (e.g., GRPO-style objectives (Shao et al., 2024)).

**Limitations.** While this additive utility model yields a simple and scalable valuation rule, it treats auxiliary datasets independently. This ignores two important aspects of the selection problem. First, auxiliary datasets may be *redundant*: multiple datasets can induce similar update directions, leading to diminishing returns when included together. Second, the usefulness of a dataset depends on the *set of other datasets* selected alongside it, rather than on its individual marginal contribution. These limitations motivate a joint valuation approach that accounts for interactions among datasets, which we develop next.

### 3.2. Kernel Mean Matching for Dataset Valuation

We now relax the additive utility assumption in Sec. 3.1 and consider a joint valuation approach that accounts for interactions among auxiliary datasets. As discussed in Sec. 2.1, the underlying post-training utility is unknown and naturally may be non-additive due to dataset interactions (Hu et al., 2024). In this section, we introduce a non-additive utility model and show that Kernel Mean Matching (KMM) yields the optimal valuation scores.

**Non-additive utility model.** We model post-training utility as a function of the selected dataset subset $\mathcal{S} \subseteq \mathcal{D}^{\text{aux}}$. Let $g(\mathcal{S}) = \sum_{i \in \mathcal{S}} g_i$ denote the *joint* update direction induced by post-training on $\mathcal{S}$. We define the utility as

$$u(\mathcal{S}) \triangleq \mathcal{L}_{\text{tar}}(\theta_0) - \mathcal{L}_{\text{tar}}(\theta_0 - \eta g(\mathcal{S})), \tag{7}$$

which is generally non-additive in $\mathcal{S}$ due to interactions among datasets through the loss landscape. This utility

captures the buyer's objective of improving target-task performance considering all datasets in $\mathcal{S}$ globally.

### 3.2.1. KMM AS GRADIENT SPACE LEAST SQUARES

To obtain a tractable optimization problem, we approximate the target loss locally around $\theta_0$. A second-order Taylor expansion yields

$$u(\mathcal{S}) = \eta\langle g_{\text{tar}}, g(\mathcal{S})\rangle - \frac{\eta^2}{2} g(\mathcal{S})^\top H_{\text{tar}}\, g(\mathcal{S}) + O\left(\eta^3 \|g(\mathcal{S})\|_2^3\right),$$
(8)

where $g_{\text{tar}} = \nabla\mathcal{L}_{\text{tar}}(\theta_0)$ and $H_{\text{tar}} = \nabla^2\mathcal{L}_{\text{tar}}(\theta_0)$.

To enable continuous optimization, we relax subset selection in Eq. (1) to weighted combinations. A subset $\mathcal{S}$ can be encoded as a binary vector $w \in \{0, 1\}^N$ with $w_i = \mathbf{1}[i \in \mathcal{S}]$, so that $g(\mathcal{S}) := \sum_{i=1}^N w_i g_i = Gw$, where $G = [g_1, \ldots, g_N]$ is the gradient matrix. Further relaxing to continuous weights $w \in \mathbb{R}^N$ and substituting into Eq. (8) defines the relaxed utility $u(w) := \eta\langle g_{\text{tar}}, Gw\rangle - \frac{\eta^2}{2}(Gw)^\top H_{\text{tar}}(Gw)$.

For clarity, we first consider the isotropic case $H_{\text{tar}} = \lambda^{-1} I_N$ with $\lambda > 0$. Maximizing $u(w)$ is equivalent to

$$\min_{w \in \mathcal{W}_k} \frac{1}{2\lambda}\|Gw\|_2^2 - \langle g_{\text{tar}}, Gw\rangle \equiv \min_{w \in \mathcal{W}_k} \|Gw - \lambda g_{\text{tar}}\|_2^2,$$
(9)

where $\mathcal{W}_k = \{w \in \mathbb{R}^N : \|w\|_1 \leq k\}$ is the convex relaxation of the cardinality constraint $|\mathcal{S}| \leq k$, and the equivalence follows from completing the square.

We can see that Eq. (9) aims to match auxiliary and target update directions. The solution $w^*$ of Eq. (9) defines dataset-level scores. As a side note, the isotropic curvature assumption is not required for the KMM formulation itself. More generally (Sec. E), any positive semidefinite local metric induces a corresponding weighted matching objective, and the same KMM structure applies.

Unlike the original KMM formulation for covariate shift (Gretton et al., 2009) (Sec. B), which enforces nonnegativity and simplex constraints so that weights define a probability distribution, we do not impose such constraints here. Dataset valuation is a signed utility signal, and allowing negative weights enables the representation of negative transfer.

### 3.2.2. KMM AS GRAM SPACE QUADRATIC PROGRAM

Let $K \in \mathbb{R}^{N \times N}$ be the Gram matrix of dataset gradients:

$$K_{ij} = \langle g_i, g_j\rangle, \qquad \beta \in \mathbb{R}^N, \ \beta_i = \langle g_i, g_{\text{tar}}\rangle.$$

Then the KMM objective Eq. (9) expands to

$$\min_{w \in \mathcal{W}_k} \frac{1}{2} w^\top K w - \lambda\beta^\top w,$$
(10)

dropping constants. The Gram matrix $K$ induces a positive semidefinite kernel via a neural-tangent-style (Jacot

et al., 2018) inner product on dataset-level gradients, and the overall problem is a convex quadratic program (QP) with a global optimum that can be solved in $O(N^3)$.

The QP form Eq. (10) makes the KMM mechanism explicit. The term $-\lambda\beta^\top w$ encourages selecting datasets whose gradients align with $g_{\text{tar}}$ (large $\beta_i$). The quadratic term $w^\top K w$ penalizes selecting datasets whose gradients are similar to each other: if $g_i$ and $g_j$ are redundant (large $K_{ij}$), then assigning weight to both increases the penalty through the cross term $\lambda w_i w_j K_{ij}$. As a result, the KMM objective jointly optimizes all weights $w_i$, so each resulting score reflects the conditional marginal contribution of dataset $i$ given the presence of other auxiliary datasets. These interaction-aware scores can be used within an additive proxy in Sec. 2.1 for budgeting decisions. In contrast, ranking by $\beta_i$ alone is the special case of Eq. (10) with $K = \mathbf{0}_N$: the redundancy penalty vanishes, so datasets are ranked only by their individual alignment with the target gradient.

**Example.** Let $\lambda = 1$. Consider a target gradient $g_{\text{tar}} = (1, 1)$ and three auxiliary gradients $g_1 = g_2 = (1, 0.1)$ and $g_3 = (0, 1)$. Their alignment scores satisfy $\langle g_1, g_{\text{tar}}\rangle = \langle g_2, g_{\text{tar}}\rangle = 1.1 > \langle g_3, g_{\text{tar}}\rangle = 1$, so a greedy alignment-based rule would select $\{g_1, g_2\}$. However, since $g_1$ and $g_2$ are identical, any linear combination of them lies in the one-dimensional subspace spanned by $(1, 0.1)$ and cannot match $g_{\text{tar}}$, regardless of the weights. In contrast, KMM's objective depends on $g_1$ and $g_2$ only through the sum of their weights, and under $\ell_1$ regularization naturally prefers sparse solutions that avoid redundancy. For instance, $w^* = (1, 0, 0.9)$ achieves zero matching error by assigning weight to only one of $\{g_1, g_2\}$ and including $g_3$ to supply the missing component needed to match the target gradient.

**Connection to Modern Portfolio Theory.** The structure of Eq. (10) admits a useful interpretation through the lens of classical mean-variance portfolio optimization (Markowitz, 1952). In the long-only setting, the analogous Markowitz objective can be written as

$$\min_{w: w \geq 0, \ \|w\|_1 = 1} \frac{1}{2} w^\top K w - \lambda\beta^\top w,$$

where $\beta$ plays the role of expected asset returns, $K$ plays the role of the asset covariance matrix, and $w$ denotes portfolio weights. This objective balances a linear reward term, $\beta^\top w$, against a quadratic risk term, $w^\top K w$.

In our setting, the same algebraic structure arises from gradient matching. The linear term $\beta^\top w$ rewards target alignment, since $\beta_i = \langle g_i, g_{\text{tar}}\rangle$ measures how well auxiliary dataset $i$ aligns with the target task. The quadratic term $w^\top K w$ penalizes redundancy, since $K_{ij} = \langle g_i, g_j\rangle$ measures pairwise similarity between auxiliary dataset gradients. Thus, assigning large weights to mutually similar datasets

increases the quadratic penalty, analogous to allocating heavily to correlated assets in a portfolio. The key difference is that, unlike long-only portfolio optimization, our valuation scores are signed and constrained by an $\ell_1$ budget, allowing negative weights to represent harmful or negatively transferring datasets.

### 3.3. Statistical and Computational Analysis

**Finite-sample estimation error of valuation scores.** Since our scores are computed from preview gradients, we quantify the error induced by finite preview samples. Let

$$w^* := \arg\min_{\|w\|_1 \leq k} \frac{1}{2} w^\top K w - \beta^\top w,$$

$$\hat{w} := \arg\min_{\|w\|_1 \leq k} \frac{1}{2} w^\top \hat{K} w - \hat{\beta}^\top w,$$

where $K_{ij} = \langle g_i, g_j \rangle$, $\beta_i = \langle g_i, g_{\text{tar}} \rangle$, and their empirical counterparts are computed using preview estimates $\hat{g}_i = m^{-1} \sum_{j=1}^m \nabla \ell_\theta(x_{ij}; \theta_0)$.

**Theorem 3.1** (Estimation error of finite-sample valuation scores). *Assume there exist universal constants $C$ and $\mu$ such that $\|\nabla \ell_\theta(x; \theta_0)\|_2 \leq C$ for all $x$, and $\lambda_{\min}(K) \geq \mu > 0$. If each preview set has size $m$, then with probability at least $1 - \delta$,*

$$\|\hat{w} - w^*\|_2 = \widetilde{O}\left(\frac{kN}{\mu\sqrt{m}}\right),$$

*where $\widetilde{O}(\cdot)$ hides logarithmic factors in $N, d, 1/\delta$, and $C$ is treated as a universal constant.*

**Remark.** This implies that the valuation scores are stable under finite preview sampling when the source-gradient Gram matrix is well-conditioned. The error decays at the usual Monte Carlo rate in the preview size $m$, scales inversely with the curvature parameter $\mu$, and grows with the number of candidate datasets $N$ and the number of selected datasets $k$ through the factor $kN$. The key idea of the proof relies on the observation that the $\arg\min$ operator of a strongly convex quadratic program is stable under perturbations of both the quadratic and linear terms, so it suffices to analyze the perturbation errors of $\|K - \hat{K}\|_2$ and $\|\beta - \hat{\beta}\|_2$ directly instead. See full proof in Sec. C.

**Computational Complexity.** Tab. 1 summarizes the computational and memory complexity of dataset valuation methods. Computing dataset-level gradients $\{g_i\}_{i=1}^N$ and the target gradient $g_{\text{tar}}$ requires $O(Nd)$ time and memory when gradients are materialized explicitly; in practice, compressed representations (e.g., random projections (Hu et al., 2025a) or low-rank adapters) can substantially reduce the effective dimensionality $d$ without affecting relative alignment. Optimizing KMM in gradient space using first-order

methods incurs $O(Nd)$ time per iteration, for a total cost of $O(TNd)$ over $T$ iterations, with $O(Nd)$ memory to store the gradient matrix. In contrast, the Gram-space formulation constructs a Gram matrix $K \in \mathbb{R}^{N \times N}$ at a cost of $O(N^2 d)$ time and $O(N^2)$ memory, followed by solving a convex quadratic program with worst-case cost $O(N^3)$. In our setting, the number of datasets $N$ is typically small (tens to hundreds), while the gradient dimensionality $d$ can be extremely large (billions), making both formulations practical.

*Table 1.* Computational and memory complexity of KMM vs. gradient methods with $N \ll d$.

| Method | Time | Memory |
|---|---|---|
| Gradient Alignment | $O(Nd)$ | $O(Nd)$ |
| KMM Gradient Space | $+ O(TNd)$ | $+ O(Nd)$ |
| KMM Gram Space | $+ O(N^2 d + N^3)$ | $+ O(N^2)$ |

## 4. Experiments

### 4.1. Dataset Valuation Experimental Setup

We evaluate dataset valuation across post-training settings, using multilingual transfer as the primary controlled benchmark and later extending to other tasks to test generalization beyond multilingual settings.

**Training.** We evaluate two post-training paradigms with LoRA (Hu et al., 2022) rank $r = 8$ by default. For multilingual SFT, we use the Aya Dataset (Singh et al., 2024) and select 26 languages that are supported by the underlying base model. For multilingual RL-GRPO, we use the GSM8K source split from Big-Math (Albalak et al., 2025) and translate problem prompts into 6 languages. We also include instruction-following experiments with math, code, and general instruction-tuning auxiliaries to test generalization beyond language-based tasks. We leave training dataset selection (Sec. F.1), hyperparameter details (Tab. 7, Tab. 8), and the effect of LoRA rank $r$ (Fig. 8) to the Appendix.

**Baselines.** We compare our gradient method against several baselines, including DataModel (Ilyas et al., 2022) and a compressed sensing (CS) variant reframed into dataset level valuation methods, as well as SOTA method GradEx-Forward-Selection (FS) and GradEx-Random-Ensemble (RE) (Li et al., 2024) applied in a highly similar dataset selection setup. DataModel is a linear regression model which requires a large amount of evaluation metrics labels from training on randomly sampled auxiliary subsets. GradEx estimates dataset values through fast linear approximations to training dynamics and random projections for dimensionality reduction, yet it depends on specific properties of the cross-entropy loss, making them only applicable to SFT. We defer more baseline method details to Sec. G. We use cosine

similarity for One-Step gradient and Task Vectors (TV) to remove the effect of gradient norms.

**Evaluation Protocol.** Given valuation scores over auxiliary datasets, according to Eq. (2), we evaluate each method by constructing auxiliary subsets of varying sizes $k$ using a greedy selection of datasets with strictly positive scores, and measuring the resulting post-training performance on the target task. To summarize performance across budget levels, we report (i) the best performance achieved over all $k$, reflecting a method's maximum attainable utility, and (ii) Borda count to aggregate rankings across $k$ values. To eliminate the confounding effect that selecting more data can trivially improve performance, we additionally introduce a *fixed-compute* evaluation protocol: the total number of optimizer updates is held constant across all values of $k$. By default, we allocate a fixed fraction of training steps to auxiliary data and distribute these steps uniformly across the selected auxiliary datasets (see Sec. H for sensitivity analysis for uniform design choice). This mixing strategy ensures that adding more auxiliary datasets redistributes, rather than increases, the auxiliary training budget. See full evaluation procedure in Sec. D.

For multilingual evaluation, we use MMMLU (Lai et al., 2023) as the target metric for supervised fine-tuning to evaluate general multilingual capability, and MGSM (Shi et al., 2023), the multilingual extension of GSM8K (Cobbe et al., 2021), as the target metric for multilingual mathematical reasoning. For instruction following, we use strict instruction-level accuracy with IFEval (Zhou et al., 2023). Evaluation task statistics are in Tab. 11.

### 4.2. Main Results

**KMM delivers strong and consistent gains across post training algorithms.** Across both GRPO (Tab. 2) and SFT (Tab. 3) evaluations, incorporating KMM consistently improves performance over the corresponding gradient-only baseline, for both the best-$k$ and average metrics. This improvement holds uniformly across all target languages, including Danish, Marathi, and Thai. Beyond these paired comparisons, KMM-enhanced methods achieve the strongest best-$k$ performance among all evaluated baselines, while their average performance is either comparable to or ranks second to the best-performing method.

**KMM achieves near-optimal performance at 2 orders of magnitude lower cost.** Fig. 2 illustrates the trade-off between computational cost and selection quality across different data valuation methods. DataModel-based approaches achieve the best average performance in Tab. 3 but incur prohibitive computational overhead due to repeated retraining on sampled subsets. This cost becomes especially pronounced for post-training settings based on RL, where a

*Table 2.* GRPO performance comparison across valuation baselines on Thai with *Qwen2.5-1.5B-Instruct* as the pretrained model.

| | \multicolumn{6}{c}{**Target = Thai**} |
|---|---|---|---|---|---|---|
| Method | $k{=}1$ | $k{=}2$ | $k{=}3$ | $k{=}4$ | $k{=}5$ | Borda |
| One Step | 35.6 | 32.8 | 39.2 | 34.8 | 31.2 | 6 |
| + KMM | 37.6 | 37.6 | 35.6 | **41.6** | 37.2 | **14** |
| TV | 36.8 | 39.6 | 37.6 | 34.4 | 34.8 | 11 |
| + KMM | 40.4 | 38.0 | 37.2 | 37.2 | 34.4 | 12 |
| Random | 30.8 | 34.8 | 32.0 | 38.4 | 36.8 | 7 |

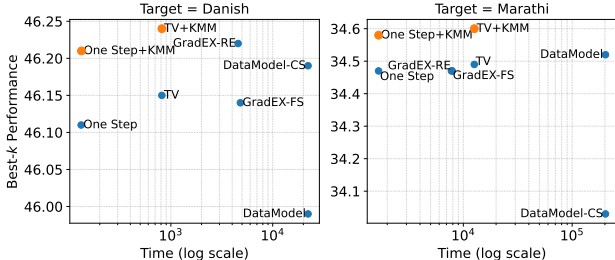

*Figure 2.* Best-$k$ performance versus wall clock time (log scale) for different data valuation methods on Danish (left) and Marathi (right). KMM-based methods are highlighted.

single training run can require substantially more time and compute. In contrast, KMM-enhanced gradient methods match this performance while achieving over $100\times$ lower runtime, and its overhead over its base gradient methods is negligible. GradEX methods partially alleviate the cost by relying on linearized models and first-order approximations, but their performance remains suboptimal. As discussed in Tab. 1, when $N \ll d$, incorporating KMM introduces only negligible additional overhead relative to gradient-only baselines. Empirically, this places KMM-based methods closer to the left-top Pareto-optimal region in Fig. 2.

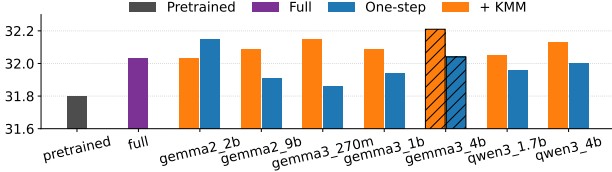

*Figure 3.* Transferability of auxiliary-task rankings across models. We evaluate whether auxiliary-dataset valuation scores computed using different pretrained models can be transferred to a model of interest, *Gemma3-4B*, for dataset selection. Bars report MMLU-Malayalam performance after fine-tuning Gemma3-4B using subsets selected by $\hat{w}_j$ derived from various source models $\theta^{\text{pub}}$ on x-axis. Full refers to SFT using full Malayalam target training set.

**KMM enables transferrable dataset valuation.** Fig. 3 demonstrates that auxiliary-task rankings computed on other models can be effectively reused for dataset selection on a

*Table 3.* SFT performance comparison across methods for Danish and Marathi as target with *Qwen3-1.7b* as the pretrained model. Blue shading denotes the best-$k$ performance per baseline, while orange shading denotes the Borda count aggregated across all $k$ values (higher is better). Fractional Borda scores arise from ties at a given $k$, where tied methods split the corresponding rank points.

| **Target = Danish** | | | | | | |
|---|---|---|---|---|---|---|
| Method | $k=5$ | $k=10$ | $k=15$ | $k=20$ | $k=25$ | Borda |
| One Step | 46.05 | 46.05 | 46.11 | 46.06 | 45.98 | 17.5 |
| + KMM | 46.21 | 46.10 | 46.15 | 46.02 | 46.08 | **29.0** |
| TV | 46.05 | 46.15 | 46.08 | 46.05 | 45.98 | 17.0 |
| + KMM | 46.05 | **46.24** | 46.10 | 46.05 | 45.98 | 20.0 |
| DataModel | 45.99 | 45.99 | 45.99 | 45.99 | 45.99 | 7.0 |
| DataModel-CS | 46.19 | 46.09 | 46.09 | 46.09 | 46.09 | 27.5 |
| GradEX-FS | 46.11 | 46.09 | 46.12 | 46.14 | 45.98 | 24.5 |
| GradEX-RE | 46.11 | 46.02 | 46.18 | 46.22 | 45.98 | 25.0 |
| Random | 45.91 | 46.04 | 46.02 | 46.15 | 45.98 | 12.5 |
| **Target = Marathi** | | | | | | |
| Method | $k=5$ | $k=10$ | $k=15$ | $k=20$ | $k=25$ | Borda |
| One Step | 34.44 | 34.42 | 34.47 | 34.40 | 34.47 | 19.0 |
| + KMM | 34.44 | 34.42 | 34.53 | 34.56 | 34.58 | 31.0 |
| TV | 34.49 | 34.26 | 34.37 | 34.39 | 34.47 | 17.5 |
| + KMM | 34.49 | 34.43 | 34.54 | 34.45 | **34.60** | **33.5** |
| DataModel | 34.51 | 34.52 | 34.52 | 34.52 | 34.52 | 33.0 |
| DataModel-CS | 34.03 | 34.03 | 34.03 | 34.03 | 34.03 | 0.0 |
| GradEX-FS | 34.38 | 34.16 | 34.32 | 34.42 | 34.47 | 13.0 |
| GradEX-RE | 34.25 | 34.18 | 34.31 | 34.42 | 34.47 | 11.5 |
| Random | 34.22 | 34.54 | 34.32 | 34.55 | 34.47 | 21.5 |

*Table 4.* Best-$k$ performance across model architectures, using instruction-following as target task. See Sec. F for task details.

| Model | TV | TV+KMM | Gain |
|---|---|---|---|
| Llama-3.2-1B | 26.26 | **27.70** | +1.44 |
| Llama-3.1-8B | 52.64 | **53.00** | +0.36 |
| Gemma-3-12B | 47.00 | **49.04** | +2.04 |
| Llama-3.1-70B | 70.10 | **71.20** | +1.10 |

tioners can run the lightweight dataset-valuation stage on a smaller model and transfer the selected datasets to fine-tune substantially larger models, supporting scalability under limited compute budgets beyond the multilingual setting.

### 4.3. Hyperparameter Analysis

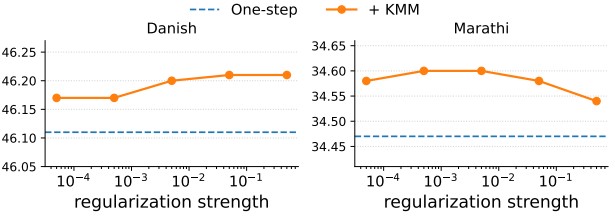

*Figure 4.* Effect of the KMM regularization parameter on downstream performance for Danish and Marathi. Solid orange curves report MMMLU metrics obtained by fine-tuning with auxiliary datasets selected using KMM under varying regularization strengths, while the dashed blue line denotes the One-Step baseline. High strength values lead to higher sparsity in the final scores.

**Robustness of KMM advantage to regularization strength.** In Fig. 4, we study KMM best-$k$ performance under the Lagrangian $\ell_1$-penalized formulation (Eq. (31)) by varying the regularization strength. KMM exhibits low sensitivity to regularization strength, achieving robust improvements over One-Step selection baseline across multiple orders of magnitude in Fig. 4, although the best regularization factor can vary between different target languages.

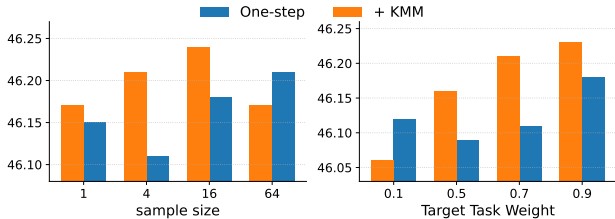

*Figure 5.* One-Step vs. KMM-based selection's MMLU-Danish best-$k$ performance under varying numbers of auxiliary task examples available for data valuation methods (left) and predefined target task weights within training batches (right).

**KMM is most effective when target signal remains informative.** In Fig. 5, KMM yields consistent gains over One-

new target model, even when the source models differ in size, version, or training paradigm (Gemma-base vs. Qwen-Instruct). As expected, rankings derived from Gemma3-4B itself yield the strongest best-$k$ performance; yet KMM consistently improves over one-step selection in all settings, and always surpasses full fine-tuning on the entire target dataset. These results imply auxiliary-task information captured by KMM is highly transferable, enabling dataset selection without direct access to the future target model.

**KMM enables scaling to larger models.** We conduct the instruction-following experiments by fine-tuning Llama-3.2-1B using instruction following as target task. Candidate auxiliary datasets consist of seven diverse instruction-tuning sources spanning math, code, and broad multi-task data. Following the cross-model transfer setup in Fig. 3, valuation scores are computed once on the 1B model and then transferred to larger models. We compare TV with TV+KMM on Llama 3.2 1B, Llama 3.1 8B, Gemma 3 12B, and Llama 3.1 70B. As shown in Tab. 4, KMM consistently improves over the TV baseline across all model sizes. Importantly, these gains are obtained even though the valuation step is performed only on the 1B model. This indicates that practi-

*Table 5.* Agreement between surrogate scores to empirical full-training utility $u_{\text{FT}}(\mathcal{S})$ over 56 Danish auxiliary subsets $\mathcal{S}$.

| Method | Spearman $\rho$ | $p$-value | Top-10 Overlap |
|---|---|---|---|
| One-Step | $-0.1746$ | 0.198 | 0/10 |
| One-Step+KMM | $-0.1265$ | 0.353 | 2/10 |
| TV | $+0.1810$ | 0.182 | 2/10 |
| **TV+KMM** | **$+0.3960$** | **0.003** | **4/10** |

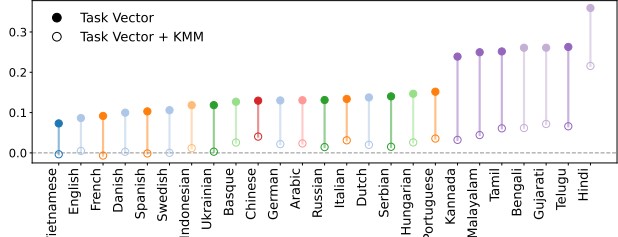

*Figure 6.* Task-Vector vs. KMM dataset valuation $w_i$ for all 25 auxiliary languages using Marathi as the target task (higher values indicate larger estimated positive contribution to target performance). Filled markers denote TV scores without reweighting, while hollow markers show scores after applying KMM. Colors indicate language subgroups from Singh et al. (2024). Languages on the x-axis are sorted by their TV scores in ascending order. KMM compresses valuation magnitudes and alters the ranking.

Step selection across a range of target data regimes. Performance advantages disappear in extreme settings where auxiliary data overwhelmingly dominates the target signal, such as when a very large number of auxiliary examples per task is used (e.g., $|\tilde{D}_i| = 64$) or when the target task weight is extremely small (e.g., 0.1). In these regimes, the combined update direction is largely determined by auxiliary gradients, leaving limited scope for selective reweighting to improve target performance. KMM provides its strongest and most reliable improvements in regimes more representative of practical scoring and selection scenarios, where auxiliary data is informative but not overwhelming and the target task retains sufficient influence to guide selection.

### 4.4. Dataset Value Scoring Analysis

**KMM estimates training utility better.** To directly test whether our surrogate rankings reflect true full-training utility, we conduct an experiment on Danish using $N = 8$ auxiliary languages, $\{\text{en}, \text{nl}, \text{sv}, \text{de}, \text{fr}, \text{es}, \text{ru}, \text{ta}\}$, and enumerate all $\binom{8}{3} = 56$ subsets of size $k = 3$. For each subset $\mathcal{S}$, we compute its additive surrogate score as $\sum_{i \in \mathcal{S}} \beta_i$ and its KMM-corrected surrogate score as $\sum_{i \in \mathcal{S}} \beta_i - \frac{1}{2} \sum_{i,j \in \mathcal{S}} K_{ij}$, using either One-Step or TV representations to define $\beta_i$ and $K_{ij}$. We compare these surrogate scores against empirical full-training utility $u_{\text{FT}}(\mathcal{S}) = \mathcal{L}_{\text{da}}^{\text{val}}(\theta_{\text{da-only}}) - \mathcal{L}_{\text{da}}^{\text{val}}(\theta_{\mathcal{S}})$, where $\mathcal{L}_{\text{da}}^{\text{val}}$ is cross-entropy on the Danish validation set and $\theta_{\mathcal{S}}$ is obtained by fine-tuning on Danish plus subset $\mathcal{S}$. Tab. 5 shows that TV+KMM achieves the strongest agreement with empirical full training utility and is the only surrogate with statistically significant positive rank correlation. The additive baselines exhibit weak agreement, while adding the KMM correction improves top-subset recovery.

**KMM reduces redundancy and promotes complementary tasks.** Fig. 6 shows TV valuation scores for auxiliary languages when Marathi is the target task. Without reweighting, TV exhibit clear grouping structure: languages that are typologically or script-wise related to Marathi (e.g., Indo-Aryan, Devanagari) receive consistently higher valuation, while more distant languages form lower-valued clusters. This grouping indicates substantial redundancy among auxiliary tasks that share similar linguistic characteristics. Applying KMM significantly alters both the magni-

tude and ordering of valuation scores. KMM systematically compresses valuation scores, suppressing weak or spurious positive signals and pushing several auxiliary tasks toward zero or negative contribution. At the same time, seemingly unrelated auxiliary tasks such as Portuguese and Italian improve in relative rank, suggesting that KMM favors tasks that provide complementary information.

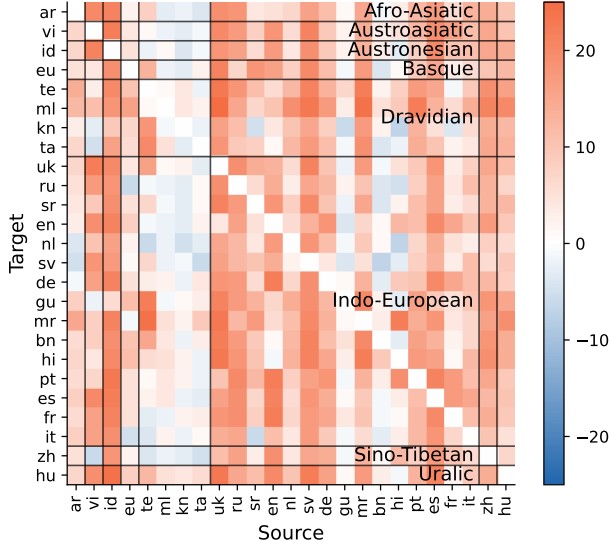

*Figure 7.* KMM-induced pairwise transfer structure across languages. Each cell shows the discrete signed transfer rank of a source language (column) for a target language (row), computed from KMM scores. Discrete scoring ranks are anchored at zero, with positive values indicating beneficial transfer, negative values indicating harmful transfer, and zero denoting neutral effect. Languages on both axes are grouped by linguistic family and ordered within each family by resource level (popularity).

**KMM reveals directional and non-trivial transfer structure.** Fig. 7 illustrates the transfer structure identified by KMM. Overall, transfer effectiveness is highly structured

*Table 6.* Effect of KMM-selected auxiliary languages on Chinese (zh) training and downstream performance. **zh-pos** augments Chinese with auxiliary languages assigned positive KMM scores, while **zh-pos-neg** additionally includes negatively scored auxiliary languages. Adding positively scored auxiliaries improves evaluation cross entropy (CE) and MMLU, but including negatively scored auxiliaries degrades evaluation and downstream performance, consistent with KMM's transfer predictions in Fig. 7.

| | $\theta_0$ | $\theta_{\text{tar}}$ | **zh-pos** | **zh-pos-neg** |
|---|---|---|---|---|
| Train CE | - | 2.8542 | 2.7132 | 1.7787 |
| Eval CE | 3.5199 | 3.3881 | 3.3819 (-0.0062) | 3.3871 (+0.0052) |
| M_MMLU | 50.44 | 50.51 | 50.64 (+0.13) | 50.49 (-0.15) |

yet asymmetric. While higher-resource languages within the same family generally provide stronger auxiliary signals than lower-resource counterparts, KMM also reveals systematic limitations, such as the consistently weak transfer from Dravidian languages, even within their own family. Importantly, KMM uncovers auxiliary relationships that are not explained by language family or data scale alone. For instance, Indonesian emerges as a strong auxiliary source for a wide range of targets, including several Indo-European languages, despite limited typological similarity. These patterns highlight KMM's ability to capture directional and complementary transfer signals beyond simple heuristics.

**KMM faithfully captures task relationships.**  Tab. 6 empirically validates that KMM scores meaningfully reflect positive and negative cross-lingual transfer. When Chinese is used as the target language, we construct a positive auxiliary set consisting of high-resource Indo-European languages (en, sv, ru, es) and a negative set drawn from the Dravidian family (te, ml, kn, ta), following pattern in Fig. 7. The results clearly support the KMM-based predictions for dataset values. Training with only positively scored languages (zh-pos) consistently improves both evaluation loss and MMMLU performance over target task only fine-tuned model $\theta_{\text{tar}}$. In contrast, augmenting the training set with negatively scored languages (zh-pos-neg) substantially reduces training loss but fails to translate into evaluation gains and degrades downstream performance.

## 5. Related Work

**Data Selection for Language Model Training.**  The performance of LLMs is highly sensitive to training data composition and quality (Hashimoto, 2021). Prior work on data selection has primarily focused on pretraining, including data filtering (Raffel et al., 2020), deduplication (Lee et al., 2022), and domain mixture optimization. More recent approaches move beyond heuristic mixing by explicitly learning domain weights. For example, DoReMi (Xie et al., 2023) casts domain reweighting as a distributionally robust optimization problem (Ben-Tal et al., 2013; Oren et al.,

2019), while RegMix (Liu et al., 2025) uses a regression-based surrogate model, inspired by DataModels (Ilyas et al., 2022), to predict downstream performance from data mixture proportions. These methods are primarily evaluated in pretraining settings and optimize next-token prediction loss.

Data selection for fine-tuning and post-training has also gained attention, with most work focusing on instance-level selection. LESS (Xia et al., 2024) selects instruction-tuning examples via low-rank gradient similarity, BIDS (Dai et al., 2025) extends this approach using iterative influence-based selection to balance multiple capabilities, and NICE (Wang et al., 2025) directly optimizes non-differentiable evaluation metrics during data selection. While effective, instance-level methods can be computationally expensive and noisy (Dai et al., 2025). In contrast, dataset- or domain-level selection for LLM post-training remains relatively underexplored. The closest related work is GradEX (Li et al., 2024), which meta-trains across tasks and uses stored projected gradients with linearity assumptions to efficiently estimate the effect of fine-tuning on arbitrary task subsets.

**Multisource and Multidomain Transfer Learning.**  Our work is closely related to multi-source domain adaptation and multi-domain training. Classic results characterize when combining multiple source domains helps or hurts, motivating careful source weighting and selection (Mansour et al., 2008; Sun et al., 2015). Related but distinct, work on multi-task learning shows that auxiliary data can benefit low-resource targets through transfer (Mueller et al., 2020; 2024), while also revealing negative transfer when irrelevant or dominant sources degrade performance (Sener & Koltun, 2018; Wang et al., 2023). Common mitigation strategies are model-side and include gradient-based methods (Yu et al., 2020), curriculum learning (Pentina et al., 2015; Bengio et al., 2009), and dynamic loss reweighting (Sener & Koltun, 2018). Overall, prior work either focuses on jointly optimizing multiple tasks or domains, or develops adaptation algorithms for combining multiple sources. In contrast, we study dataset-level selection in a multisource setting with a *single* downstream objective, framing multisource training as a data-centric transfer problem.

## 6. Conclusion

We formulate dataset valuation as a budget-aware post-training problem and show that independent gradient-based scores fail to capture redundancy among auxiliary datasets. We propose a scalable KMM method that yields actionable valuations for dataset selection and weighting. Experiments across diverse post-training settings demonstrate consistent improvements over existing baselines with minimal overhead, establishing dataset valuation as a practical tool for cost-aware LLM development.

## Acknowledgment

Siqi Zeng and Han Zhao are supported by a Meta grant and an NSF CAREER Award No. 2442290. The authors would like to thank Yifei He and Meitong Liu for their discussion throughout the development of this work. This work was conducted as part of a research collaboration between UIUC and Meta. The views, opinions, and conclusions expressed in this paper are solely those of the authors and do not necessarily reflect the views or official positions of Meta Platforms, Inc.

## Impact Statement

This work aims to improve the efficiency and transparency of dataset selection during post-training of large language models. By providing a principled, redundancy-aware valuation signal, the proposed method can help practitioners make more informed decisions under fixed compute or data budgets, potentially reducing redundant training and data acquisition. We emphasize that the resulting scores are task-, model-, and metric-dependent, and do not represent intrinsic or universal dataset value. As with other data valuation methods, inappropriate use outside the intended context, such as for pricing or exclusion decisions without human oversight, may lead to misleading conclusions.

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

# A. Additional Related Work

**Data Market Modeling and Valuation-Aware Transactions.**   Several works study data marketplaces and data acquisition as economic or algorithmic problems for classic machine learning settings. Early efforts model two-sided data markets and pricing mechanisms for training data, focusing on incentive compatibility, quality uncertainty, or signaling between buyers and sellers (Agarwal et al., 2019; Chen et al., 2022). Recent work begins to link data valuation to pricing in markets for foundation model training data. Chen et al. (2025) introduce the DAM challenge within DataPerf (Mazumder et al., 2023), framing data acquisition as a benchmarked problem with simulated marketplaces and budget constraints, and empirically showing that gradient-based budgeting strategies can perform well across different market designs. Zhang et al. (2025) propose a valuation-driven pricing framework for LLM data markets, modeling strategic interactions between buyers and sellers and showing that valuation-aligned pricing improves both fairness and model performance. While closely related in spirit, this line of work primarily studies pricing mechanisms and equilibrium properties, rather than improving valuation signals to guide downstream budgeting decisions.

# B. Kernel Mean Matching for Covariate Shift

KMM was originally proposed as a principled method for handling covariate shift, where the training distribution differs from the test distribution while the conditional label distribution remains unchanged (Gretton et al., 2009).

Let $\mathcal{X}$ denote the input space and let $\Phi : \mathcal{X} \to \mathcal{H}$ be a feature map into a reproducing kernel Hilbert space (RKHS) $\mathcal{H}$ with associated kernel

$$k(x, x') = \langle \Phi(x), \Phi(x') \rangle_{\mathcal{H}}.$$

For any probability distribution $\mathcal{P}$ on $\mathcal{X}$, its kernel mean embedding is defined as

$$\mu(P) := \mathbb{E}_{x \sim \mathcal{P}}[\Phi(x)] \in \mathcal{H}. \tag{11}$$

Consider a training distribution $\mathcal{P}_{\text{tr}}$ and a test distribution $\mathcal{P}_{\text{te}}$. The goal of KMM is to estimate an importance weighting function $\beta : \mathcal{X} \to \mathbb{R}_+$ such that the feature-space mean of the reweighted training distribution matches that of the test distribution:

$$\mathbb{E}_{x \sim \mathcal{P}_{\text{tr}}}[\beta(x)\Phi(x)] \approx \mu(\mathcal{P}_{\text{te}}).$$

This leads to the following optimization problem:

$$\min_{\beta} \quad \|\mu(\mathcal{P}_{\text{te}}) - \mathbb{E}_{x \sim \mathcal{P}_{\text{tr}}}[\beta(x)\Phi(x)]\|_{\mathcal{H}}^2 \tag{12}$$

$$\text{s.t.} \quad \beta(x) \geq 0, \qquad \mathbb{E}_{x \sim P_{\text{tr}}}[\beta(x)] = 1. \tag{13}$$

The constraints ensure that $\beta(x)\mathcal{P}_{\text{tr}}(x)$ defines a valid probability distribution.

Under the assumption that $\mathcal{P}_{\text{te}}$ is absolutely continuous with respect to $\mathcal{P}_{\text{tr}}$ and that the kernel is universal, the optimal solution to (12),(13) recovers the true density ratio,

$$\beta^*(x) = \frac{\mathcal{P}_{\text{te}}(x)}{\mathcal{P}_{\text{tr}}(x)},$$

and achieves zero objective value. In practice, expectations are replaced by empirical averages, yielding a finite-dimensional convex quadratic program that can be efficiently solved.

# C. Finite Sample Bounds on Valuation Scores

Since our solution obtained through the preview gradients is a noisy version of the true gradient vector, now we want to explore how much error is introduced due to this estimation.

For $\mathcal{W}_k = \{w \in \mathbb{R}^N : \|w\|_1 \leq k\}$, assume $w$ is the solution for the following constrained objective

$$w^* = \arg \min_{w \in \mathcal{W}_k} J(w) := \frac{1}{2}w^\top K w - \beta^\top w, \tag{14}$$

where $K_{ij} = \langle g_i, g_j \rangle$ and $\beta_i = \langle g_i, g_{\text{tar}} \rangle$ defined by the true expected gradient (*e.g.*, $g_i = \nabla \mathcal{L}_i(\theta_0) = \mathbb{E}_{x \sim \mathcal{P}_i}[\nabla \ell_\theta(x; \theta_0)]$). Due to limited data access during data valuation stage, we solve the following objective instead with solution $\hat{w}$:

$$\hat{w} = \arg\min_{w \in \mathcal{W}_k} \hat{J}(w) := \frac{1}{2} w^\top \hat{K} w - \hat{\beta}^\top w, \tag{15}$$

where $\hat{K}_{ij} = \langle \hat{g}_i, \hat{g}_j \rangle$ and $\hat{\beta}_i = \langle \hat{g}_i, g_{\text{tar}} \rangle$ defined by the empirical gradient collected on dataset preview with $\hat{g}_i = \frac{1}{m} \sum_{j=1}^{m} \nabla \ell_\theta(x_{ij}; \theta_0)$ for *i.i.d.* samples $x_{ij}$ sampled from preview $\tilde{D}_i$.

Compared to Eq. (10), without loss of generality, we assume $\lambda = 1$.

**Assumption C.1** (Bounded Gradients). Assume $\|\nabla \ell_\theta(x; \theta_0)\|_2 \leq C, \forall x$, then it implies $\|\hat{g}_i\|_2 \leq C, \|g_i\|_2 \leq C, \|g_{\text{tar}}\| \leq C$.

**Assumption C.2** (Gram Matrix PD). We assume the true Gram matrix $K$ is strictly positive definite with a minimum eigenvalue bounded below by a constant $\mu > 0$: i.e., $\lambda_{\min}(K) \geq \mu$.

**Proof Sketch.** By the convexity argument in the residual decomposition lemma, the solution error $\|\hat{w} - w^*\|_2$ is reduced to controlling the empirical errors in $\hat{K}$ and $\hat{\beta}$. These errors come from replacing the true gradients $g_i$ by the preview gradients $\hat{g}_i$. Matrix Bernstein controls the corresponding gradient estimation error with probability at least $1 - \delta$. Plugging this concentration event into the residual decomposition bound proves the theorem.

**Lemma C.3** (Residual Decomposition). *Assume $\lambda_{\min}(K) \geq \mu$. Then,*

$$\|\hat{w} - w^*\|_2 \leq \frac{1}{\mu} \|\nabla \hat{J}(\hat{w}) - \nabla J(\hat{w})\|_2 \leq \frac{1}{\mu} \left( \|(\hat{K} - K)\hat{w}\|_2 + \|\hat{\beta} - \beta\|_2 \right). \tag{16}$$

*Proof.* Since $\lambda_{\min}(K) \geq \mu$, we know $J$ is strongly convex, so

$$J(\hat{w}) - J(w^*) \geq \langle \nabla J(w^*), \hat{w} - w^* \rangle + \frac{\mu}{2} \|\hat{w} - w^*\|_2^2,$$

$$J(w^*) - J(\hat{w}) \geq \langle \nabla J(\hat{w}), w^* - \hat{w} \rangle + \frac{\mu}{2} \|\hat{w} - w^*\|_2^2.$$

Sum them up, we have,

$$\langle \nabla J(\hat{w}) - \nabla J(w^*), \hat{w} - w^* \rangle \geq \mu \|\hat{w} - w^*\|_2^2. \tag{17}$$

Since Eq. (14) and Eq. (15) are both convex problems, by first order optimality condition, we have,

$$\left\langle \nabla \hat{J}(\hat{w}), \hat{w} - w^* \right\rangle \leq 0, \tag{18}$$

and similarly,

$$\langle \nabla J(w^*), \hat{w} - w^* \rangle \geq 0. \tag{19}$$

Therefore, we can rewrite Eq. (17) as:

$$\begin{aligned}
\|\hat{w} - w^*\|_2^2 &\leq \frac{1}{\mu} \langle \nabla J(\hat{w}) - \nabla J(w^*), \hat{w} - w^* \rangle \\
&= \frac{1}{\mu} \left( \langle \nabla J(\hat{w}), \hat{w} - w^* \rangle - \langle \nabla J(w^*), \hat{w} - w^* \rangle \right) \\
&\leq \frac{1}{\mu} \langle \nabla J(\hat{w}), \hat{w} - w^* \rangle && \text{(Eq. (19))} \\
&= \frac{1}{\mu} \left( \left\langle \nabla J(\hat{w}) - \nabla \hat{J}(\hat{w}), \hat{w} - w^* \right\rangle + \left\langle \nabla \hat{J}(\hat{w}), \hat{w} - w^* \right\rangle \right) \\
&\leq \frac{1}{\mu} \left\langle \nabla J(\hat{w}) - \nabla \hat{J}(\hat{w}), \hat{w} - w^* \right\rangle && \text{(Eq. (18))} \\
&\leq \frac{1}{\mu} \|\nabla \hat{J}(\hat{w}) - \nabla J(\hat{w})\|_2 \cdot \|\hat{w} - w^*\|_2.
\end{aligned}$$

Canceling the solution gap term from both sides, we have

$$\|\hat{w} - w^*\|_2 \le \frac{1}{\mu} \|\nabla \hat{J}(\hat{w}) - \nabla J(\hat{w})\|_2. \tag{20}$$

Now we plug in gradient values. Since both Eq. (14) and Eq. (15) share the same constraints, $w^*, \hat{w}$ are all feasible for 2 objectives. Then,

$$\begin{aligned}
\|\nabla \hat{J}(\hat{w}) - \nabla J(\hat{w})\|_2 &= \|(\hat{K}\hat{w} - \hat{\beta}) - (K\hat{w} - \beta)\|_2 \\
&\le \|(\hat{K} - K)\hat{w}\|_2 + \|\hat{\beta} - \beta\|_2.
\end{aligned}$$

$\square$

We can first decompose the residual terms in Eq. (16).

Let

$$G = \begin{bmatrix} g_1^\top \\ \vdots \\ g_N^\top \end{bmatrix} \in \mathbb{R}^{N \times d}, \qquad \hat{G} = \begin{bmatrix} \hat{g}_1^\top \\ \vdots \\ \hat{g}_N^\top \end{bmatrix} \in \mathbb{R}^{N \times d}.$$

Define

$$\Delta := \hat{G} - G.$$

Then by definition of $K$ and $\hat{K}$, we have

$$\begin{aligned}
\|\hat{K} - K\|_2 &= \|\hat{G}\hat{G}^\top - GG^\top\|_2 \\
&= \|(G + \Delta)(G + \Delta)^\top - GG^\top\|_2 \\
&= \|G\Delta^\top + \Delta G^\top + \Delta\Delta^\top\|_2 \\
&\le \|G\Delta^\top\|_2 + \|\Delta G^\top\|_2 + \|\Delta\Delta^\top\|_2 \\
&\le \|G\|_2\|\Delta\|_2 + \|\Delta\|_2\|G\|_2 + \|\Delta\|_2^2 \\
&= 2\|G\|_2\|\Delta\|_2 + \|\Delta\|_2^2. \tag{21}
\end{aligned}$$

Similarly, by definition of $\beta$ and $\hat{\beta}$, we have

$$\begin{aligned}
\|\hat{\beta} - \beta\|_2 &= \|\hat{G}g_{\text{tar}} - Gg_{\text{tar}}\|_2 \\
&= \|(\hat{G} - G)g_{\text{tar}}\|_2 \\
&= \|\Delta g_{\text{tar}}\|_2 \\
&\le \|\Delta\|_2\|g_{\text{tar}}\|_2. \tag{22}
\end{aligned}$$

Therefore, to control the solution gap in Eq. (16), we need to find the upper bound for $\|\Delta\|_2$.

**Lemma C.4** (Matrix Bernstein (Tropp, 2015) Theorem 6.1.1). *Let $X_1, \ldots, X_M$ be independent mean-zero random matrices in $\mathbb{R}^{N \times d}$. Suppose that there exists $L > 0$ such that*

$$\|X_k\|_2 \le L \qquad \text{almost surely for all } k \in [M]. \tag{23}$$

*Define the variance parameter*

$$\sigma^2 := \max \left\{ \left\| \sum_{k=1}^{M} \mathbb{E}[X_k X_k^\top] \right\|_2, \left\| \sum_{k=1}^{M} \mathbb{E}[X_k^\top X_k] \right\|_2 \right\}. \tag{24}$$

*Then, for any $\delta \in (0, 1)$, with probability at least $1 - \delta$,*

$$\left\| \sum_{k=1}^{M} X_k \right\|_2 \le \sqrt{2\sigma^2 \log \frac{N + d}{\delta}} + \frac{2L}{3} \log \frac{N + d}{\delta}. \tag{25}$$

**Lemma C.5** (Concentration of Preview Gradient Matrix). *For each $i \in [N]$ and $j \in [m]$, define*

$$Z_{ij} := \nabla \ell_\theta(x_{ij}; \theta_0) - g_i \in \mathbb{R}^d.$$

*Under Assumption C.1, for any $\delta \in (0, 1)$, with probability at least $1 - \delta$,*

$$\|\Delta\|_2 \le \varepsilon_\Delta(\delta), \tag{26}$$

*where*

$$\varepsilon_\Delta(\delta) := \sqrt{2v \log \frac{N + d}{\delta}} + \frac{4C}{3m} \log \frac{N + d}{\delta}, \tag{27}$$

*and*

$$v := \frac{1}{m} \max \left\{ \max_{i \in [N]} \mathbb{E}\|Z_{ij}\|_2^2, \left\| \sum_{i=1}^{N} \mathbb{E}[Z_{ij} Z_{ij}^\top] \right\|_2 \right\}. \tag{28}$$

*In particular, since $\|Z_{ij}\|_2 \le 2C$, one may take*

$$v \le \frac{4NC^2}{m}.$$

*Proof.* For each $i \in [N]$ and $j \in [m]$, define

$$X_{ij} := \frac{1}{m} e_i Z_{ij}^\top \in \mathbb{R}^{N \times d},$$

where $e_i$ is the $i$-th standard basis vector in $\mathbb{R}^N$. Then,

$$\sum_{i=1}^{N} \sum_{j=1}^{m} X_{ij} = \sum_{i=1}^{N} \sum_{j=1}^{m} \frac{1}{m} e_i \left( \nabla \ell_\theta(x_{ij}; \theta_0) - g_i \right)^\top = \hat{G} - G = \Delta.$$

Also, $\mathbb{E}[X_{ij}] = 0$. By Assumption C.1,

$$\|Z_{ij}\|_2 \le \|\nabla \ell_\theta(x_{ij}; \theta_0)\|_2 + \|g_i\|_2 \le 2C.$$

Therefore,

$$\|X_{ij}\|_2 = \frac{1}{m} \|e_i Z_{ij}^\top\|_2 = \frac{1}{m} \|Z_{ij}\|_2 \le \frac{2C}{m}.$$

Now we compute the variance parameter. Since

$$X_{ij} X_{ij}^\top = \frac{1}{m^2} e_i Z_{ij}^\top Z_{ij} e_i^\top = \frac{1}{m^2} \|Z_{ij}\|_2^2 e_i e_i^\top,$$

we have

$$\left\| \sum_{i=1}^{N} \sum_{j=1}^{m} \mathbb{E}[X_{ij} X_{ij}^\top] \right\|_2 = \left\| \frac{1}{m} \sum_{i=1}^{N} \mathbb{E}\|Z_{ij}\|_2^2 e_i e_i^\top \right\|_2 = \frac{1}{m} \max_{i \in [N]} \mathbb{E}\|Z_{ij}\|_2^2.$$

Similarly,

$$X_{ij}^\top X_{ij} = \frac{1}{m^2} Z_{ij} Z_{ij}^\top,$$

so

$$\left\| \sum_{i=1}^{N} \sum_{j=1}^{m} \mathbb{E}[X_{ij}^\top X_{ij}] \right\|_2 = \frac{1}{m} \left\| \sum_{i=1}^{N} \mathbb{E}[Z_{ij} Z_{ij}^\top] \right\|_2.$$

Thus the variance parameter is exactly Eq. (28).

Applying Lem. C.4 to $\Delta = \sum_{i=1}^{N} \sum_{j=1}^{m} X_{ij}$ gives, with probability at least $1 - \delta$,

$$\|\Delta\|_2 \le \sqrt{2v \log \frac{N+d}{\delta}} + \frac{2}{3} \cdot \frac{2C}{m} \log \frac{N+d}{\delta}$$

$$= \sqrt{2v \log \frac{N+d}{\delta}} + \frac{4C}{3m} \log \frac{N+d}{\delta} = \varepsilon_\Delta(\delta).$$

Finally, since $\|Z_{ij}\|_2 \le 2C$,

$$v \le \frac{1}{m} \max\left\{4C^2, 4NC^2\right\} = \frac{4NC^2}{m}.$$

This proves the lemma. $\qquad\qquad\square$

**Theorem C.6** (High-Probability Stability Bound). *Under Assumption C.1 and Assumption C.2, with probability at least $1 - \delta$,*

$$\|\hat{w} - w^*\|_2 \le \frac{\varepsilon_\Delta(\delta)}{\mu} \left[2k\sqrt{N}C + k\varepsilon_\Delta(\delta) + C\right]. \tag{29}$$

*Proof.* By Lem. C.3,

$$\|\hat{w} - w^*\|_2 \le \frac{1}{\mu}\left(\|(\hat{K} - K)\hat{w}\|_2 + \|\hat{\beta} - \beta\|_2\right)$$

$$\le \frac{1}{\mu}\left(\|(\hat{K} - K)\|_2\|\hat{w}\|_2 + \|\hat{\beta} - \beta\|_2\right)$$

$$\le \frac{1}{\mu}\left(k\|(\hat{K} - K)\|_2 + \|\hat{\beta} - \beta\|_2\right) \qquad (\hat{w} \in \mathcal{W}_k)$$

By Lem. C.5,

$$\|\Delta\|_2 \le \varepsilon_\Delta(\delta).$$

By Eq. (21),

$$\|\hat{K} - K\|_2 \le 2\|G\|_2\varepsilon_\Delta(\delta) + \varepsilon_\Delta^2(\delta).$$

By Eq. (22) and Assumption C.1,

$$\|\hat{\beta} - \beta\|_2 \le \varepsilon_\Delta(\delta)\|g_{\text{tar}}\|_2 \le C\varepsilon_\Delta(\delta).$$

Combining above gives us

$$\|\hat{w} - w^*\|_2 \le \frac{1}{\mu}\left[k\left(2\|G\|_2\varepsilon_\Delta(\delta) + \varepsilon_\Delta^2(\delta)\right) + C\varepsilon_\Delta(\delta)\right]$$

$$= \frac{\varepsilon_\Delta(\delta)}{\mu}\left[2k\|G\|_2 + k\varepsilon_\Delta(\delta) + C\right].$$

Finally, by Assumption C.1, $\|g_i\|_2 \le C$ for all $i \in [N]$. Thus,

$$\|G\|_2 \le \|G\|_F = \left(\sum_{i=1}^{N} \|g_i\|_2^2\right)^{1/2} \le \sqrt{N}C.$$

This gives us the final upper bound as follows

$$\|\hat{w} - w^*\|_2 \le \frac{\varepsilon_\Delta(\delta)}{\mu}\left[2k\sqrt{N}C + k\varepsilon_\Delta(\delta) + C\right].$$

$\qquad\qquad\square$

**High-level rate.** We now record the leading-order dependence of the bound on $N, m, d, \delta$, and $\mu$. Treat $C$ as a universal constant, and define

$$L_{N,d,\delta} := \log\left(\frac{N+d}{\delta}\right).$$

By Lem. C.5 and $v \le 4NC^2/m$, we have

$$\varepsilon_\Delta(\delta) = O\left(C\sqrt{\frac{NL_{N,d,\delta}}{m}} + C\frac{L_{N,d,\delta}}{m}\right).$$

Substituting this bound into

$$\|\hat{w} - w^*\|_2 \le \frac{\varepsilon_\Delta(\delta)}{\mu}\left[2k\sqrt{N}C + k\varepsilon_\Delta(\delta) + C\right],$$

and writing $L = L_{N,d,\delta}$, gives

$$\|\hat{w} - w^*\|_2 = O\left(\frac{C^2}{\mu}\left[kN\sqrt{\frac{L}{m}} + k\sqrt{N}\frac{L}{m} + k\frac{NL}{m} + k\frac{\sqrt{N}L^{3/2}}{m^{3/2}} + k\frac{L^2}{m^2} + \sqrt{\frac{NL}{m}} + \frac{L}{m}\right]\right).$$

In the sample-size regime $m \gtrsim L$, higher-order terms in $L/m$ are dominated by the leading terms. Since $N \ge 1$ in our setting, we have

$$k\sqrt{N}\frac{L}{m}, \quad k\frac{NL}{m}, \quad k\frac{\sqrt{N}L^{3/2}}{m^{3/2}}, \quad k\frac{L^2}{m^2} = O\left(kN\sqrt{\frac{L}{m}}\right).$$

Similarly,

$$\frac{L}{m} = O\left(\sqrt{\frac{L}{m}}\right) = O\left(\sqrt{\frac{NL}{m}}\right).$$

Therefore, restoring $L = L_{N,d,\delta}$, we obtain

$$\|\hat{w} - w^*\|_2 = O\left(\frac{C^2}{\mu}\left[kN\sqrt{\frac{L_{N,d,\delta}}{m}} + \sqrt{\frac{NL_{N,d,\delta}}{m}}\right]\right).$$

Equivalently, treating $C$ as constant and treating $k$ as a fixed budget, the $\sqrt{N}$ term is dominated by the $kN$ term. Thus,

$$\|\hat{w} - w^*\|_2 = \widetilde{O}\left(\frac{kN}{\mu\sqrt{m}}\right),$$

where $\widetilde{O}(\cdot)$ hides logarithmic factors in $N, d, 1/\delta$.

## D. Details of Evaluation Protocol

Alg. 1 specifies the evaluation protocol used to assess a given dataset ranking via full multi-step fine-tuning. Note that it is not used during the computation of valuation scores. For the fixed computed training step, we include the key training hyperparameters for SFT in Tab. 7 and GRPO in Tab. 8.

**Hardware.** All experiments were run on NVIDIA A6000 GPUs (48GB VRAM). SFT training used a single A6000 GPU, with lm-eval-harness (Gao et al., 2024) evaluation performed on a separate dedicated A6000 GPU. GRPO training used six A6000 GPUs, with evaluation and the vLLM (Kwon et al., 2023) inference server each assigned to their own dedicated A6000 GPU to avoid resource contention.

## E. General Curvature Derivation of KMM

The derivation in Sec. 3.2.1 assumes isotropic curvature for simplicity. Here we show that the resulting KMM formulation generalizes naturally to arbitrary positive semidefinite local curvature.

*Table 7.* SFT training hyperparameters for Qwen/Qwen3-1.7B.

| Hyperparameter | Value |
|---|---|
| Max sequence length ($L$) | 1024 |
| Max steps | 25 |
| Batch size (per-device train) | 8 |
| Gradient accumulation steps | 2 |
| Optimizer | `adamw_torch` |
| Learning rate | $1.0 \times 10^{-4}$ |
| LR scheduler | `cosine` |
| Warmup | ratio = 0.03 |
| Max grad norm | 1.0 |
| LoRA rank $r$ | 8 |
| LoRA $\alpha$ | 16 |
| LoRA dropout | 0.05 |
| LoRA target modules | $\{$`q_proj,k_proj,v_proj`$\}$ |
| Precision | bf16 |

*Table 8.* GRPO training hyperparameters for Qwen/Qwen2.5-1.5B-Instruct.

| Hyperparameter | Value |
|---|---|
| Max prompt length | 256 |
| Max completion length | 1024 |
| Max steps | 500 |
| Batch size (per-device train) | 8 |
| Gradient accumulation steps | 1 |
| Optimizer | `adamw_torch` |
| Learning rate | $1.0 \times 10^{-6}$ |
| LR scheduler | `cosine` |
| Warmup | ratio = 0.04 |
| Max grad norm | 1.0 |
| Num generations | 8 |
| Temperature | 0.6 |
| Top-$p$ | 0.95 |
| KL coeff. ($\beta$) | 0.005 |
| LoRA rank $r$ | 8 |
| LoRA $\alpha$ | 32 |
| LoRA dropout | 0.05 |
| LoRA target modules | $\{$`q,k,v,o,up,down,gate`$\}$`_proj` |
| Precision | bf16 |

---

**Algorithm 1** Budget-Constrained Evaluation of Dataset Valuation Methods

---

**Require:** Target dataset $D_{\text{tar}}$; auxiliary datasets $\{D_i\}_{i=1}^N$; valuation weights $\{w_i\}_{i=1}^N$; candidate thresholds $\mathcal{K}$; step budget
  $T$; target-mixing ratio $\{\rho_k\}_{k\in\mathcal{K}}$
**Ensure:** Fixed-compute performances $\{P_k^{\text{fix}}\}_{k\in\mathcal{K}}$; best fixed-compute score $P^{\star,\text{fix}}$
 1: Sort auxiliary indices by weight in descending order: $w_{\pi(1)} \geq w_{\pi(2)} \geq \cdots \geq w_{\pi(N)}$
 2: Let $\mathcal{I}^+ \leftarrow \{\, i \mid w_i > 0 \,\}$
 3: **for** each $k \in \mathcal{K}$ **do**
 4:    $\mathcal{S}_k \leftarrow \emptyset$
 5:    **for** $j = 1$ to $N$ **do**
 6:       **if** $\pi(j) \in \mathcal{I}^+$ **then**
 7:          $\mathcal{S}_k \leftarrow \mathcal{S}_k \cup \{\pi(j)\}$
 8:       **end if**
 9:       **if** $|\mathcal{S}_k| = k$ **then**
10:          **break**
11:       **end if**
12:    **end for**
13:    $D_{\text{aux}}^{(k)} \leftarrow \bigcup_{i\in\mathcal{S}_k} D_i$
14:    // Fixed-compute training
15:    Initialize model parameters $\theta^{\text{fix}} \leftarrow \theta_0$
16:    **for** $t = 1$ to $T$ **do**
17:       // one optimizer update
18:       Sample minibatch $b_{\text{tar}} \sim D_{\text{tar}}$
19:       Sample minibatch $b_{\text{aux}} \sim D_{\text{aux}}^{(k)}$
20:       **if** $\text{Bernoulli}(\rho_k) = 1$ **then**
21:          $b \leftarrow b_{\text{tar}}$
22:       **else**
23:          $b \leftarrow b_{\text{aux}}$
24:       **end if**
25:       $\theta^{\text{fix}} \leftarrow \text{Update}(\theta^{\text{fix}}; b)$
26:    **end for**
27:    $P_k^{\text{fix}} \leftarrow \text{Eval}(\theta^{\text{fix}}; D_{\text{tar}}^{\text{val/test}})$
28: **end for**
29: $P^{\star,\text{fix}} \leftarrow \max_{k\in\mathcal{K}} P_k^{\text{fix}}$

---

Starting from the second-order approximation

$$\mathcal{L}_{\text{tar}}(\theta_0 - \eta g(w)) \approx \mathcal{L}_{\text{tar}}(\theta_0) - \eta\langle g_{\text{tar}}, g(w)\rangle + \frac{\eta^2}{2} g(w)^\top H_{\text{tar}} g(w),$$

where $H_{\text{tar}} \succeq 0$, minimizing the predicted loss is equivalent to

$$\min_{w\in\mathcal{W}_B} \frac{1}{2} g(w)^\top H_{\text{tar}} g(w) - \langle g_{\text{tar}}, g(w)\rangle.$$

Since $H_{\text{tar}} \succeq 0$, let $H_{\text{tar}}^{1/2}$ denote its symmetric square root and $H_{\text{tar}}^{\dagger/2}$ the square root of its Moore–Penrose pseudoinverse.
Completing the square yields

$$\min_{w\in\mathcal{W}_B} \left\| H_{\text{tar}}^{1/2} g(w) - H_{\text{tar}}^{\dagger/2} g_{\text{tar}} \right\|_2^2,$$

up to additive constants. Defining transformed update vectors $\tilde{g}_i = H_{\text{tar}}^{1/2} g_i$ and $\tilde{g}_{\text{tar}} = H_{\text{tar}}^{\dagger/2} g_{\text{tar}}$, this recovers *the same
least-squares matching structure* as Eq. (9) in a transformed feature space. The isotropic case $H_{\text{tar}} = \lambda^{-1} I$ corresponds to
a global rescaling of this metric and is therefore a special case of the above formulation. In practice, the local metric is
generally unknown and need not be estimated; the KMM formulation only requires a consistent representation of update
directions, and the instantiation used in the main text constitutes one practical choice that performs well empirically.

*Table 9.* Best-$k$ results comparing the Euclidean KMM metric with a diagonal empirical-Fisher curvature approximation. Each entry reports test metric/CE loss.

| Method | Danish | Marathi |
|---|---|---|
| One-Step+KMM (Euclidean) | 46.21 / 2.6202 | 34.58 / 1.0875 |
| One-Step+KMM (Diag Fisher) | 46.15 / 2.6301 | 34.58 / 1.0948 |
| TV+KMM (Euclidean) | 46.24 / 2.6231 | 34.60 / 1.0880 |
| TV+KMM (Diag Fisher) | 46.06 / 2.6277 | 34.58 / 1.0931 |

We also evaluated a concrete non-isotropic instantiation using a diagonal empirical-Fisher approximation. Specifically, we set

$$H_{\text{tar}} \approx M = \text{diag}(\widehat{F}), \qquad \widehat{F} = \frac{1}{n} \sum_{i=1}^{n} g_i g_i^\top, \tag{30}$$

and applied the same KMM quadratic program after transforming each update direction as $\tilde{g}_i = M^{1/2} g_i$. This gives a practical realization of the generalized PSD-curvature formulation without requiring a full Hessian estimate. The resulting comparison is shown in Tab. 9.

Empirically, the diagonal Fisher variant is slightly but consistently worse than the Euclidean baseline. Thus, in this setting, introducing anisotropic diagonal curvature does not improve subset quality. We interpret this as an empirical bias–variance tradeoff: while the generalized formulation is operational and admits non-isotropic curvature, this particular diagonal approximation appears too noisy in the high-dimensional regime considered here. Consequently, we use the Euclidean metric as the preferred practical instantiation in the main experiments.

# F. Datasets

**Why multilingual tasks.**   We choose multilingual datasets for our experiments because they naturally reflect practical post-training scenarios in which target-task supervision is limited and auxiliary data from related or unrelated languages must be leveraged to improve performance. Multilingual settings provide a controlled yet realistic testbed for studying positive and negative transfer under data scarcity, while allowing systematic variation in resource levels across tasks. Moreover, our formulation is model- and algorithm-agnostic, making it straightforward to extend to a wide range of post-training methods.

**Instruction-following Task Experiments.**   For the instruction-following experiments, we use allenai/tulu-3-sft-personas-instruction-following (Lambert et al., 2024) as the target task and evaluate pretrained (not instruction-tuned) models using strict instruction-level accuracy with IFEval (Zhou et al., 2023). The auxiliary candidate pool consists of seven diverse instruction-tuning sources: allenai/tulu-3-sft-personas-math-filtered, allenai/tulu-3-sft-personas-math-grade-filtered, allenai/tulu-3-sft-personas-algebra, allenai/tulu-3-sft-personas-code, SurgeGlobal/Evol-Instruct (Dissanayake et al., 2024), HuggingFaceTB/smol-smoltalk (Allal et al., 2025), and teknium/openhermes (Teknium, 2023). These datasets cover math, algebraic reasoning, code, general instruction following, and broad multi-task instruction tuning. We sample 3K examples from each auxiliary dataset before computing valuation scores.

## F.1. Train

### F.1.1. SFT

Aya (Singh et al., 2024) is a multilingual benchmark that categorizes languages by resource level (e.g., High/Mid/Low) in practice. We select the language set in Tab. 10 because these languages are reported as supported by Qwen3 models (per the Qwen3 technical report (Yang et al., 2025)), and for Gemma3 (Team et al., 2025) we additionally verified that the tokenizer includes the Unicode characters needed to represent text in that script, so the model can tokenize and generate text in that language without falling back to unknown or degenerate tokens. Within this set, we emphasize two target scenarios for studying transfer from high-resource data to low-resource performance: (i) **Danish** has an unusually small number of training data points (#DP), simulating a scarce-supervision setting even though it is labeld as Mid-resource by Aya; and (ii) **Marathi** is explicitly labeled Low-resource by Aya. Together, these targets let us simulate two realistic regimes where high-resource auxiliary data may improve performance when target training information is limited.

*Table 10.* Languages used in our experiments with Aya resource labels and the number of training data points (#DP). Danish (`da`) and Marathi (`mr`) are the target languages in our transfer experiments. In SFT experiments, once we fix the target task, we use the rest of 25 tasks in this table as auxiliary tasks.

| Lang. Code | Language | Script | Family | Subgrouping | Resources | # Training DP |
|---|---|---|---|---|---|---|
| ar | Arabic | Arabic | Afro-Asiatic | Semitic | High | 4995 |
| eu | Basque | Latin | Basque | – | High | 939 |
| bn | Bengali | Bengali | Indo-European | Indo-Aryan | Mid | 1534 |
| zh | Simplified Chinese | Han | Sino-Tibetan | Sinitic | High | 4909 |
| da | Danish | Latin | Indo-European | Germanic | Mid | 97 |
| nl | Dutch | Latin | Indo-European | Germanic | High | 1733 |
| en | English | Latin | Indo-European | Germanic | High | 3944 |
| fr | French | Latin | Indo-European | Italic | High | 1422 |
| de | German | Latin | Indo-European | Germanic | High | 241 |
| gu | Gujarati | Gujarati | Indo-European | Indo-Aryan | Low | 3989 |
| hi | Hindi | Devanagari | Indo-European | Indo-Aryan | High | 1153 |
| hu | Hungarian | Latin | Uralic | – | High | 98 |
| id | Indonesian | Latin | Austronesian | Malayo-Polynesian | Mid | 786 |
| it | Italian | Latin | Indo-European | Italic | High | 738 |
| kn | Kannada | Kannada | Dravidian | South Dravidian | Low | 334 |
| ml | Malayalam | Malayalam | Dravidian | South Dravidian | Low | 1749 |
| mr | Marathi | Devanagari | Indo-European | Indo-Aryan | Low | 3545 |
| pt | Portuguese | Latin | Indo-European | Italic | High | 8997 |
| ru | Russian | Cyrillic | Indo-European | Balto-Slavic | High | 423 |
| sr | Serbian | Cyrillic | Indo-European | Balto-Slavic | High | 152 |
| es | Spanish | Latin | Indo-European | Italic | High | 3854 |
| sv | Swedish | Latin | Indo-European | Germanic | High | 1310 |
| ta | Tamil | Tamil | Dravidian | South Dravidian | Mid | 14133 |
| te | Telugu | Telugu | Dravidian | South Dravidian | Low | 8439 |
| uk | Ukrainian | Cyrillic | Indo-European | Balto-Slavic | Mid | 522 |
| vi | Vietnamese | Latin | Austroasiatic | Vietic | High | 8676 |

### F.1.2. GRPO

Our GRPO experiments use six multilingual tasks: Thai (`th`), Chinese (`zh`), Vietnamese (`vi`), English (`en`), Hindi (`hi`), and Spanish (`es`), each consisting of 1822 samples. We fix the data source across languages by selecting 1822 English GSM8K math problems from the Big-Math-RL-Verified-Processed dataset (`open-r1/Big-Math-RL-Verified-Processed`, (Albalak et al., 2025; Hugging Face, 2025)). Non-English tasks are created by automatically translating the English problems using the Google Translate Python package. We designate **Thai** as the target task and treat the remaining languages as auxiliary tasks, motivated by its Mid-resource classification, its inclusion in multilingual math evaluation benchmarks, and its support in Qwen 2.5 (Qwen et al., 2025).

### F.2. Evaluation

We conduct all evaluations using the `lm-eval` (Gao et al., 2024) framework (Tab. 11). For SFT experiments, we evaluate on M_MMLU_da (Danish) and M_MMLU_mr (Marathi), containing 13206 and 12313 samples respectively, and report accuracy as the primary metric. For GRPO experiments, we evaluate on the Thai test split of MGSM, which consists of 250 examples. Following prior multilingual math evaluation practice, we report THE exact-match-flexible-extract metric to account for minor formatting variations in model outputs while preserving answer correctness.

Besides, all non-gradient-based baselines require validation signals to estimate dataset utility. To evaluate their robustness when the true target metric is expensive or unavailable, we may use proxy metrics during valuation. For SFT, we compute validation CE loss on the Dolly-machine-translated (Conover et al., 2023) split from the Aya's evaluation suite. For RL, we directly evaluate on the target test metric.

*Table 11.* Evaluation tasks and statistics used in our experiments.

| Stage | Task | Language | # Test Samples | Metric |
|---|---|---|---|---|
| SFT | M_MMLU_da (Lai et al., 2023) | Danish | 13206 | Accuracy |
| SFT | M_MMLU_mr (Lai et al., 2023) | Marathi | 12313 | Accuracy |
| GRPO | MGSM (Shi et al., 2023) | Thai | 250 | Exact Match (Flexible-Extract) |

## G. Data Valuation Baselines

### G.1. Gradient-based Methods

#### G.1.1. KMM

Instead of enforcing the $\ell_1$ budget constraint $w \in \mathcal{W}_k$, we can incorporate sparsity directly into the objective via an $\ell_1$ penalty with regularization strength $\gamma > 0$. Using the Gram matrix $K_{ij} = \langle g_i, g_j \rangle$ and alignment vector $\beta_i = \langle g_i, g_{\text{tar}} \rangle$, we solve the unconstrained kernelized problem in Gram matrix form (leaving the quadratic term implicit can lead to a larger canonical form and significantly slower solve times for convex solvers):

$$w^* = \arg\min_{w \in \mathbb{R}^N} \frac{1}{2} w^\top K w - \beta^\top w + \gamma \|w\|_1, \tag{31}$$

which is the Lagrangian relaxation of Eq. (10) with $\|w\|_1 \leq k$ and $\lambda = 1$. Equivalently, Eq. (31) can be viewed as a kernelized LASSO: the quadratic term $w^\top K w$ penalizes selecting redundant datasets, while the linear term $-\beta^\top w$ rewards alignment with the target gradient, and the $\ell_1$ penalty promotes sparse valuation weights. As $\gamma$ increases, the solution becomes sparser; as $\gamma \to 0$, the solution approaches the minimum-norm matching solution in Gram space.

We solve the $\ell_1$-regularized KMM objective as a convex quadratic program using CVXPY (Diamond & Boyd, 2016; Agrawal et al., 2018), with OSQP (Stellato et al., 2020) as the default solver.

### G.2. DataModel

Datamodel methods cast dataset valuation as a linear regression problem over subset-existence features (Ilyas et al., 2022). Specifically, we sample $m$ auxiliary dataset index subsets $\{\mathcal{S}_r\}_{r=1}^m$, where each $\mathcal{S}_r \subseteq \{1, \ldots, N\}$ specifies a set of auxiliary datasets to include in post-training. For each subset $\mathcal{S}_r$, we train a post-trained model on $D^{\text{tar}} \cup \bigcup_{i \in \mathcal{S}_r} D_i$ and record a scalar target evaluation outcome $y_r \in \mathbb{R}$ (e.g., negative loss or accuracy). We then fit a sparse linear model

$$y \approx Aw, \qquad A \in \mathbb{R}^{m \times N}, \ \ y \in \mathbb{R}^m, \ \ w \in \mathbb{R}^N, \tag{32}$$

where $w_i$ is the estimated valuation score of dataset $D_i$ for the fixed target task. Following prior work, we encourage sparse solutions by solving a LASSO problem

$$\hat{w} = \arg\min_{w \in \mathbb{R}^N} \|Aw - y\|_2^2 + \alpha \|w\|_1, \tag{33}$$

which matches the standard datamodel formulation with $\ell_1$ regularization (Ilyas et al., 2022).

**Uniform Datamodel.** The uniform baseline uses a binary subset-existence design matrix $A^{\text{uni}} \in \{0, 1\}^{m \times N}$, where each row corresponds to one sampled auxiliary subset. Concretely, for each row $r \in [m]$, we sample a subset $\mathcal{S}_r \subseteq [N]$ of fixed size $\text{round}(\rho N)$ (for inclusion fraction $\rho \in (0, 1)$), and set

$$A^{\text{uni}}_{r,i} = \mathbb{I}\{i \in \mathcal{S}_r\}. \tag{34}$$

We then train a model $\theta_{\mathcal{S}_r} = \text{FT}(\theta_0, \{D^{\text{tar}}\} \cup \{D_i : i \in \mathcal{S}_r\})$ and evaluate it on the target task to obtain a scalar measurement

$$y_r^{\text{uni}} = \text{Eval}(\theta_{\mathcal{S}_r}; D^{\text{tar}}). \tag{35}$$

Finally, we fit $\hat{w}$ by plugging $(A^{\text{uni}}, y^{\text{uni}})$ into Eq. (33). This is the standard datamodel regression with a binary mask matrix indicating which training datasets were present in each counterfactual training subset (Ilyas et al., 2022).

**Compressed-Sensing Datamodel.** The compressed-sensing variant replaces the binary design with a sparse random $\{-1, 0, +1\}$ construction tailored to compressive sensing (Achlioptas, 2003). For each row $r \in [m]$ and dataset index $i \in [N]$, we draw i.i.d.

$$A_{r,i}^{\mathrm{cs}} \;=\; c \cdot \xi_{r,i}, \qquad c = \sqrt{\frac{3}{N}}, \qquad \xi_{r,i} \in \{-1, 0, +1\}, \tag{36}$$

with $\Pr(\xi_{r,i} = +1) = \frac{1}{6}, \Pr(\xi_{r,i} = 0) = \frac{2}{3}$, and $\Pr(\xi_{r,i} = -1) = \frac{1}{6}$. This is the Achlioptas-type sparse projection used in the compressive-sensing datamodel baseline (Achlioptas, 2003).

Because $A^{\mathrm{cs}}$ has signed entries, each row $r$ is implemented via two auxiliary subsets whose difference realizes the signed measurement. Define

$$\mathcal{S}_{r,1} \;=\; \{i \in [N] : \xi_{r,i} \in \{0, +1\}\}, \qquad \mathcal{S}_{r,2} \;=\; \{i \in [N] : \xi_{r,i} \in \{0, -1\}\}. \tag{37}$$

Intuitively, indices with $\xi_{r,i} = 0$ appear in both subsets (intersection), while $\xi_{r,i} = +1$ appear only in $\mathcal{S}_{r,1}$ and $\xi_{r,i} = -1$ appear only in $\mathcal{S}_{r,2}$. We train two models, $\theta_{\mathcal{S}_{r,1}} = \mathrm{FT}(\theta_0, \{D^{\mathrm{tar}}\} \cup \{D_i : i \in \mathcal{S}_{r,1}\})$ and $\theta_{\mathcal{S}_{r,2}} = \mathrm{FT}(\theta_0, \{D^{\mathrm{tar}}\} \cup \{D_i : i \in \mathcal{S}_{r,2}\})$, evaluate both on the target task, and form a difference measurement

$$y_r^{\mathrm{cs}} \;=\; c \left( \mathrm{Eval}(\theta_{\mathcal{S}_{r,1}}; D^{\mathrm{tar}}) \;-\; \mathrm{Eval}(\theta_{\mathcal{S}_{r,2}}; D^{\mathrm{tar}}) \right). \tag{38}$$

Finally, we fit $\hat{w}$ by solving Eq. (33) with $(A^{\mathrm{cs}}, y^{\mathrm{cs}})$. This procedure matches the compressive-sensing adaptation of the datamodel regression, where the signed design is realized by paired subset trainings and the response is a rescaled metric difference.

**Key difference.** Both methods estimate per-dataset valuation scores by solving the same sparse linear regression Eq. (33); the only difference is the design matrix construction: binary subset-existence masks $A^{\mathrm{uni}} \in \{0, 1\}^{m \times N}$ for the uniform baseline versus a sparse signed random design $A^{\mathrm{cs}}$ (implemented via paired subsets) for the compressed-sensing baseline.

### G.3. GradEx

We next describe GradEx, a dataset valuation approach that estimates the effect of auxiliary datasets on post-training performance without repeatedly fine-tuning the model. GradEx is adapted from the first-order approximation framework of Li et al. (2024) and is closely related to TRAK (Park et al., 2023). The key idea is to exploit a local linearization of supervised fine-tuning objectives around a shared meta-initialization.

**Meta-initialization via multitask post-training.** Let $\theta_0 \in \mathbb{R}^d$ denote the pretrained model. GradEx first performs multitask post-training on the union of all available datasets,

$$\theta^\star \;=\; \mathrm{FT}\big(\theta_0, \{D^{\mathrm{tar}}\} \cup \mathcal{D}^{\mathrm{aux}}\big), \tag{39}$$

yielding a meta-initialization $\theta^\star$. Models fine-tuned on subsets of auxiliary datasets are assumed to remain close (in parameter space) to $\theta^\star$, which enables accurate local approximations of fine-tuning dynamics.

**First-order reduction via linearization of the model output.** Recall that in Sec. 3.1 $\ell(x; \theta)$ denotes the per-example loss of a model with parameters $\theta$ on input $x$. In supervised fine-tuning, $x$ implicitly includes the label (or target sequence), and we equivalently write $z = (x, y)$ when it is helpful to make the label explicit. GradEx, following TRAK (Park et al., 2023), linearizes a carefully chosen scalar model output function instead of loss values, which allows the resulting surrogate optimization to remain convex.

We now specialize to multiclass supervised cross-entropy objectives, where $\ell(z; \theta) = L(z; \theta)$ with

$$L(z; \theta) \;:=\; -\sum_{c=1}^{C} \mathbb{I}[y = c] \log p(c \mid x; \theta) \;=\; -\log p(y \mid x; \theta), \qquad z = (x, y). \tag{40}$$

As shown in TRAK, the key insight is to define a scalar model output as the log-odds of the correct class,

$$f(z; \theta) \;:=\; \log\left( \frac{p(y \mid x; \theta)}{1 - p(y \mid x; \theta)} \right), \tag{41}$$

where $p(y \mid x; \theta)$ is the softmax probability assigned to the correct label. A crucial property of this choice is that it allows the multi-class cross-entropy loss to be rewritten exactly as a logistic loss in terms of the scalar output:

$$L(z; \theta) = \log(1 + \exp(-f(z; \theta))). \tag{42}$$

GradEx then applies a first-order Taylor expansion to the scalar output $f(z; \theta)$ around a reference parameter vector $\theta^\star$,

$$f(z; \theta) \approx f(z; \theta^\star) + \nabla_\theta f(z; \theta^\star)^\top (\theta - \theta^\star), \tag{43}$$

and substitutes this approximation into Eq. (42).

Substituting the linearized output Eq. (43) into Eq. (42) yields the surrogate objective

$$\min_\theta \sum_{z_i \in \mathcal{S}} \log\Big(1 + \exp\big(-\nabla_\theta f(z_i; \theta^\star)^\top (\theta - \theta^\star)\big)\Big), \tag{44}$$

which is exactly the objective of binary logistic regression with feature vectors $\nabla_\theta f(z_i; \theta^\star)$ and parameter vector $\theta - \theta^\star$.

Importantly, this reduction treats the entire multi-class problem as a single logistic regression problem, rather than decomposing it into multiple one-vs-rest subproblems. Here the binary logistic surrogate does not introduce explicit $\pm 1$ labels: each example contributes a positive logistic loss corresponding to increasing the log-odds of its correct class, with class competition handled implicitly by the softmax.

To make this surrogate computationally tractable in large models, GradEx applies a random projection to the gradient features $\nabla_\theta f(z; \theta^\star)$, preserving inner products by the Johnson-Lindenstrauss lemma (Johnson et al., 1984).

Given any auxiliary subset $\mathcal{S} \subseteq \mathcal{D}^{\text{aux}}$, GradEx aggregates the projected features across all examples $z$ contained in $\mathcal{S}$ and solves the corresponding surrogate to produce a fast estimate of the target-task evaluation metric, denoted $\widehat{\text{Eval}}(\mathcal{S})$. This estimate approximates the true post-training performance $\text{Eval}(\theta_\mathcal{S}; D^{\text{tar}})$, where $\theta_\mathcal{S} = \text{FT}(\theta_0, \{D^{\text{tar}}\} \cup \mathcal{S})$, while avoiding explicit fine-tuning on $\mathcal{S}$.

**Extension to generative modeling and limitations.** For autoregressive language modeling under SFT, the sequence-level cross-entropy decomposes into a sum of token-level multi-class cross-entropies. Applying Eq. (41)-Eq. (44) at each output position and averaging the resulting gradients yields a valid GradEx estimator for generative SFT settings.

This reduction is specific to supervised cross-entropy objectives: it relies on defining the log-odds output $f(z; \theta)$ so that cross-entropy can be rewritten exactly as a logistic loss. Reinforcement learning objectives (e.g., policy-gradient or actor–critic losses) are defined over trajectories and stochastic returns and do not admit an analogous log-odds reparameterization into a convex logistic surrogate.

**GradEx-FS (forward selection).** GradEx-FS integrates $\widehat{\text{Eval}}(\cdot)$ into greedy forward selection. Starting from $\mathcal{S} = \emptyset$, at each step it evaluates $\widehat{\text{Eval}}(\mathcal{S} \cup \{D_i\})$ for all remaining datasets $D_i \in \mathcal{D}^{\text{aux}}$ and adds the dataset yielding the largest estimated improvement. The procedure stops when no remaining dataset improves the estimate. Replacing full fine-tuning with first-order estimation makes forward selection scalable to many datasets.

**GradEx-RE (random ensemble).** GradEx-RE instead samples random auxiliary subsets $\{\mathcal{S}_r\}_{r=1}^m$ and computes $\widehat{\text{Eval}}(\mathcal{S}_r)$ for each. Each dataset $D_i$ is then assigned an aggregate valuation weight

$$w_i^{\text{RE}} = \frac{1}{|\{r : D_i \in \mathcal{S}_r\}|} \sum_{r: D_i \in \mathcal{S}_r} \widehat{\text{Eval}}(\mathcal{S}_r), \tag{45}$$

i.e., the average estimated target performance across all sampled subsets in which $D_i$ appears. Datasets are ranked by $w_i^{\text{RE}}$ and selected by thresholding or taking the top entries.

### G.4. Baseline hyperparameters

For all baselines, we report the best performance obtained over a set of method-specific hyperparameters.

*Table 12.* Popularity-based auxiliary task sampling weights. Higher values indicate a higher probability of being sampled within the auxiliary portion of each batch.

| Language | en | zh | es | de | fr | ru | pt | it | ar | hi | id | nl | sv | bn | da | eu | gu | hu | kn | ml | mr | sr | ta | te | uk | vi |
|---|---|---|---|---|---|---|---|---|---|---|---|---|---|---|---|---|---|---|---|---|---|---|---|---|---|---|
| **Weight** | 2.0 | 2.0 | 1.5 | 1.5 | 1.5 | 1.5 | 1.2 | 1.2 | 1.2 | 1.2 | 1.0 | 1.0 | 1.0 | 0.8 | 0.8 | 0.8 | 0.8 | 0.8 | 0.8 | 0.8 | 0.8 | 0.8 | 0.8 | 0.8 | 0.8 | 0.8 |

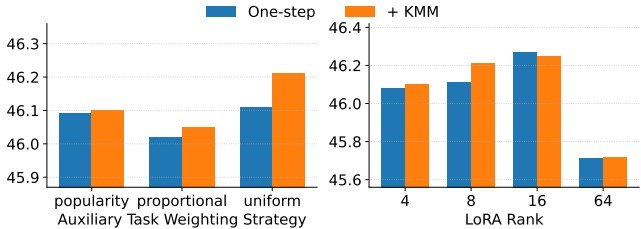

*Figure 8.* One-step vs. KMM-based selection best-$k$ performance under different auxiliary data weighting strategies (left) and LoRA adapter ranks (right) for Danish as target task. Popularity weights auxiliary tasks based on resource availability to reflect realistic language corpus composition, proportional corresponds to the original Aya training distribution, and uniform denotes equal weighting across tasks (our default).

For TV baselines, we vary the number of fine-tuning steps used to construct task vectors, considering half, equal to, and double the default training steps used in the target fine-tuning configuration in Sec. D.

For KMM, we tune the $\ell_1$ regularization strength in the penalized formulation, selecting $\gamma \in \{5 \times 10^{-2}, 5 \times 10^{-4}\}$.

For GradEx, we use the random-ensemble (RE) variant with $10N$ randomly sampled subsets, where $N$ is the total number of auxiliary datasets, following protocol in Li et al. (2024). For both GradEx-FS and GradEx-RE, we fix the random projection dimension to 100.

For DataModel baselines, for a fair comparison with GradEx, we also run a total of $10N$ post-training runs. For the compressed-sensing variant, each measurement is implemented via paired subset trainings, resulting in half as many rows in the design matrix for the same total training budget.

## H. Additional Experiments

**Robustness of KMM to Auxiliary Task Distribution Weighting.** We consider three auxiliary task distribution weighting strategies: *uniform*, *proportional*, and *popularity*-based. Under the uniform strategy, the auxiliary budget $1 - \rho_k$ is evenly allocated across all auxiliary tasks. In contrast, the proportional strategy allocates auxiliary sampling probability in proportion to each task's training data size (Tab. 10), while the popularity-based strategy assigns weights according to real-world language popularity and resource categorization. Sampling weights (Tab. 12) are normalized within the auxiliary pool, with larger weights corresponding to a higher probability of being sampled in each training batch.

In Fig. 8, across all auxiliary data weighting strategies, KMM consistently outperforms one-step selection. Notably, this holds even when the auxiliary data distribution closely reflects realistic corpus composition, indicating that KMM does not rely on a particular weighting heuristic to be effective.

*Table 13.* Best-$k$ accuracy on Danish using non-uniform softmax weighting over the selected top-$k$ datasets. For each method, we evaluate $k \in \{5, 10, 15, 20, 25\}$ and report the best result.

| Method | One Step | One Step+KMM | TV | TV+KMM | DataModel | DataModel-CS | GradEX-FS | GradEX-RE | Random |
|---|---|---|---|---|---|---|---|---|---|
| Best-$k$ Acc. | 45.96 | **46.12** | 45.99 | 46.07 | 45.90 | 46.03 | 46.05 | 46.07 | 45.96 |

We also evaluate whether using the learned scores as non-uniform mixture weights improves downstream performance. For each method and each $k \in \{5, 10, 15, 20, 25\}$, we first select the top-$k$ datasets according to the corresponding scores, discard all datasets outside the top-$k$ set, and then assign mixture weights by applying a softmax over the selected top-$k$ scores. This allows higher-scoring selected datasets to receive larger training weights while retaining the same top-$k$ selection structure. The best result across $k$ for each method on Danish is reported in Tab. 13. Compared with the uniform-weighting

results in Tab. 3, non-uniform softmax weighting leads to lower absolute performance for all methods in this setting. Nevertheless, One Step+KMM remains the best-performing method, and KMM continues to improve over its corresponding gradient-based baseline.

**Effect of LoRA Adapter Capacity on KMM.** KMM's advantage over one-step selection persists across a range of LoRA adapter ranks, demonstrating robustness to changes in fine-tuning capacity. In 3 out of 4 comparisons of Fig. 8, KMM yields results stronger than its one-step baselines for most ranks, indicating that its benefits are not tied to a specific parameter budget.

*Table 14.* Best-$k$ accuracy on RoBERTa using 13 auxiliary NLU tasks. This evaluates whether KMM generalizes beyond decoder-only LLMs to an encoder-only architecture.

| Method | One Step | One Step+KMM | TV | TV+KMM | DataModel | DataModel-CS | GradEX-FS | GradEX-RE |
|---|---|---|---|---|---|---|---|---|
| Best-$k$ Acc. | 96.22 | 96.67 | 96.67 | **96.90** | 95.76 | 96.56 | 95.87 | 95.64 |

**Results for Encoder-based Architecture.** To assess whether the proposed KMM correction transfers beyond decoder-only LLMs, we additionally evaluate it with RoBERTa, an encoder-only architecture, using 13 auxiliary NLU tasks: CoLA (Warstadt et al., 2018), CR (Hu & Liu, 2004), MNLI (Williams et al., 2017), MPQA (Wiebe et al., 2005), MR (Pang & Lee, 2004), MRPC (Dolan & Brockett, 2005), QNLI (Wang, 2018), QQP (Sharma et al., 2019), RTE (Wang, 2018), SNLI (Bowman et al., 2015), SST-5 (Socher et al., 2013), SUBJ (Pang & Lee, 2004), and TREC (Li & Roth, 2002). The results are reported in Tab. 14. KMM consistently improves its corresponding gradient-based baseline. Moreover, Task Vector+KMM achieves the best overall accuracy among all evaluated methods. These results suggest that the proposed matching objective is not tied to decoder-only architectures.

**Multi-seed Results.** We additionally report multi-seed results with five random seeds for both the multilingual SFT and multilingual GRPO settings. The SFT results are shown in Tab. 15. Although the absolute gains in the multilingual SFT setting are modest, the ranking across methods is stable: KMM-based variants consistently rank among the strongest methods. In particular, TV+KMM achieves the best accuracy on both Danish and Marathi. For Danish, the confidence intervals of TV and TV+KMM do not overlap, suggesting that the improvement is statistically meaningful. The multilingual GRPO results are shown in Tab. 16. These results further confirm the single-run findings.

*Table 15.* Multi-seed best-$k$ results on the multilingual SFT setting over five random seeds.

| Method | Danish | Marathi |
|---|---|---|
| One Step | $46.10 \pm 0.02$ | $34.48 \pm 0.08$ |
| One Step+KMM | $46.18 \pm 0.03$ | $34.60 \pm 0.08$ |
| TV | $46.26 \pm 0.06$ | $34.37 \pm 0.09$ |
| TV+KMM | $\mathbf{46.38 \pm 0.07}$ | $\mathbf{34.76 \pm 0.11}$ |
| DataModel | $45.99 \pm 0.05$ | $34.43 \pm 0.06$ |
| DataModel-CS | $46.10 \pm 0.06$ | $33.96 \pm 0.13$ |
| GradEX-FS | $46.15 \pm 0.01$ | $34.41 \pm 0.04$ |
| GradEX-RE | $46.23 \pm 0.01$ | $34.53 \pm 0.08$ |
| Random | $46.11 \pm 0.02$ | $34.46 \pm 0.09$ |

*Table 16.* Multi-seed best-$k$ results on the multilingual GRPO setting over five random seeds.

| Method | Best-$k$ Acc. (%) |
|---|---|
| One Step | $37.80 \pm 1.41$ |
| One Step+KMM | $\mathbf{42.06 \pm 0.87}$ |
| TV | $38.62 \pm 0.78$ |
| TV+KMM | $40.27 \pm 0.66$ |
| Random | $37.59 \pm 0.92$ |

**Dependency on Linearization.** We next evaluate whether the advantage of KMM persists when training deviates further from the linearization point. Using the same valuation scores, we train on Danish with $0.25\times$, $0.5\times$, $2\times$, and $4\times$ the default number of training steps, where larger budgets correspond to updates farther from the local approximation. The results are shown in Tab. 17.

*Table 17.* Best-$k$ accuracy on Danish under different training budgets. The same valuation scores are used across budgets, and larger budgets move training farther from the local Taylor approximation.

| Training Steps | Random | One Step | One Step+KMM |
|---|---|---|---|
| $0.25\times$ | 45.69 | 45.87 | **45.93** |
| $0.5\times$ | 45.80 | 46.01 | **46.08** |
| $2\times$ | 45.75 | 45.97 | **45.99** |
| $4\times$ | 45.56 | 45.87 | **46.01** |

We note that these results are not directly comparable to the main results in Tab. 3: there, we tune the number of epochs to maximize performance uniformly across all methods, whereas here we intentionally use non-optimal training budgets to stress-test the local approximation. Consequently, the absolute scores are slightly lower. Nevertheless, One Step+KMM consistently outperforms One Step across all training budgets, indicating that the benefit of KMM is robust even as training moves farther from the local second-order Taylor approximation.

**Scaling to Larger Candidate Pools.** We evaluate scalability on the instruction-following task by varying the number of candidate data groups, $N \in \{100, 500, 1000\}$. To make the scaling experiment reflect selection among meaningfully different data sources, we construct candidate groups using embedding-based clustering rather than random splits. Specifically, we encode all training examples with sentence embeddings (Reimers & Gurevych, 2019), cluster them in embedding space, and treat each cluster as one candidate dataset. This produces candidate groups that are internally semantically coherent while remaining diverse across groups. We then compute One Step scores for each candidate group and apply KMM on top of these scores to select the final subset. Downstream performance is measured by fine-tuning Llama 3.2 3B on the selected data and evaluating best-$k$ instruction-following accuracy. As shown in Tab. 18, One Step+KMM consistently improves over One Step across all tested candidate-pool sizes.

*Table 18.* Downstream best-$k$ performance on Llama 3.2 3B for the instruction-following task using clustered candidate splits. Candidate groups are constructed by clustering examples with sentence embeddings.

| $N$ | One Step | One Step+KMM | Gain |
|---|---|---|---|
| 100 | 42.33 | **44.72** | +2.39 |
| 500 | 45.32 | **45.56** | +0.24 |
| 1000 | 44.36 | **47.24** | +2.88 |

*Table 19.* Wall-clock time for One Step score computation and KMM quadratic-program solving across different numbers of candidate groups.

| Method | $N = 100$ | $N = 500$ | $N = 1000$ |
|---|---|---|---|
| One Step | 8.1 min | 40.5 min | 88.6 min |
| KMM | 0.9–1.4 min | 1.0 min | 2.4–2.5 min |

We also measure the computational cost of the two main stages: computing One Step scores and solving the KMM quadratic program. As reported in Tab. 19, One Step scales approximately linearly with the number of candidate groups. In contrast, the KMM optimization adds only a small CPU-side overhead. One Step uses $5.5$GB of GPU memory, constant across $N$, since candidate groups are processed sequentially for model-based gradient computation. KMM is CPU-only and uses $0$GB GPU memory. The CPU memory footprint of KMM remains modest at the tested scales, as shown in Tab. 20.

Overall, these measurements show that the dominant cost is computing the underlying gradient-based scores, while the additional KMM step is negligible at the tested scales. Although generic quadratic programming can have unfavorable worst-case scaling in $N$, the observed wall-clock time and memory footprint indicate that KMM is practical for substantially

*Table 20.* Peak CPU memory usage of the KMM quadratic program.

| $N$ | KMM CPU Peak |
|-----|--------------|
| 100 | 34 MB |
| 500 | 21 MB |
| 1000 | 137 MB |

larger candidate pools. Extrapolating from these measurements, KMM itself could scale to $N = 100{,}000$ on a small lab server, requiring roughly 16 hours for the QP solve and about 40GB of CPU memory to load the Gram matrix.

