# OpenReview forum: "Convex Dataset Valuation for Post-Training"
_ICML.cc/2026/Conference — ICML 2026 regular_

### Official Review · Reviewer_F6g4 · 2026-03-10

**Soundness:** 3
**Presentation:** 3
**Significance:** 3
**Originality:** 3
**Overall Recommendation:** 4
**Confidence:** 3

**Summary:**

This paper studies dataset valuation for LLM post-training in settings where a developer has a target task with a limited amount of data but also has access to multiple auxiliary datasets that may improve downstream performance. The authors formulate dataset valuation as a subset selection problem, where each dataset represents a discrete acquisition decision instead of individual data points. They first show that simple gradient alignment scores provide a basic signal but fail to account for redundancy between datasets. To address this, the paper proposes a convex dataset valuation method based on kernel mean matching (KMM) in gradient space, which jointly models alignment with the target task and interactions among datasets. The resulting formulation becomes a convex quadratic optimization problem, enabling efficient computation of dataset weights. Experiments in multilingual LLM post-training demonstrate that the method improves target-task performance compared with existing gradient-based or DataModel baselines while remaining computationally efficient.

**Compliance With Llm Reviewing Policy:**

Affirmed.

**Final Justification:**

Most of my concerns are resolved. Additional experiment results show KMM's potential on different model, tasks and dataset size.

**Key Questions For Authors:**

1. How sensitive are the resulting dataset rankings to the local Taylor approximation, and have the authors evaluated whether rankings remain stable when the training trajectory deviates significantly from the linearization point? For instance, by reporting ranking correlations against actual fine-tuning outcomes across varying numbers of training steps.
2. The valuation scores depend on dataset-level gradient estimates. How sensitive are the results to gradient estimation noise, such as different batch sizes, subsets of data, or random seeds?
3. The proposed approach requires constructing a Gram matrix over datasets and solving a convex quadratic program. How well does the method scale when the number of candidate datasets becomes very large, since one key assumption is that both the target and auxiliary datasets are not large and KMM has a complexity of O(N^3)?
4. How well does the proposed method generalize to post-training settings beyond multilingual tasks, such as multimodal tasks, where dataset interactions may differ fundamentally?

**Limitations:**

Yes, there is a limitation section. Please see the weaknesses above for my suggestions.

**Strengths And Weaknesses:**

Strengths
1. The paper formulates dataset valuation specifically for post-training of LLMs, which is a realistic and important setting in emerging data marketplaces.
2. The formulation leads to a convex optimization problem, which provides computational tractability and avoids expensive retraining or combinatorial search used in many valuation approaches.
3. The experiments evaluate the approach in multilingual post-training scenarios and show improvements over several valuation baselines.

Weaknesses
1. The approach relies on local approximations of the loss landscape (e.g., Taylor expansion and gradient representations), which may not accurately capture the complex training dynamics of large-scale models on large datasets.
2. The method assumes access to dataset-level gradients or task vectors, which may be expensive or difficult to obtain for very large datasets or models.
3. The experiment results show improvements, but they are limited compared with baselines, even with random selection.
4. The connections between KMM and Portfolio Theory are not fully discussed. Either provide a thorough analysis of the similarities and inspirations between these two equations or do not include them. The current paragraph looks somewhat distracting from the convex optimization perspective.

---

> ### Author Rebuttal · Authors · 2026-03-31
>
> **W1. Local approximation limitations.**
>
> > **Response:** We evaluate whether KMM's advantage holds when training deviates further from the linearization point. Using the same valuation scores, we train on Danish with x0.25, x0.5, x2, x4 of the default training steps (higher steps = further from local approximation):
> >
> > |Steps|Random|One Step|One Step+KMM|
> > |---|---|---|---|
> > |x0.25|45.69|45.87|**45.93**|
> > |x0.5|45.80|46.01|**46.08**|
> > |x2|45.75|45.97|**45.99**|
> > |x4|45.56|45.87|**46.01**|
> >
> > Note: in Table 3, we tuned epochs to maximize performance universally for all methods, removing any disadvantage due to suboptimal hyperparameters. Here, scores are slightly lower as we use non-optimal training budgets. KMM consistently outperforms One Step across all training budgets, showing that the advantage from our method is robust even as training moves further from the local 2nd-order Taylor approximation.
>
> **W2. Gradient computation cost and sensitivity to gradient estimation noise.**
>
> > **Response (cost):** Please see our response to Reviewer arAi W3 for scaling experiments up to N=1000 with extrapolation to N=100K. Gradient computation itself is lightweight: we use only a small sample size (default = 4) per dataset. We use LoRA for gradient computation, and for very large-scale settings, further random projection methods can additionally reduce dimensionality and compute requirements.
> >
> > **Response (sensitivity):** As shown in Figure 5, our method maintains robust advantage across different sample sizes >= 1. Varying the sample size is essentially using different subsets of data for Monte Carlo gradient estimation, and KMM's advantage remains stable. We also provide a formal finite-preview approximation bound in our response to Reviewer cMDm W2, showing the dominant error decays as $O(m^{-1/2})$. For random seed sensitivity: please see our multi-seed results (5 seeds, 95% CI) in our response to Reviewer arAi W1-W2, where KMM variants consistently rank highest across seeds.
>
> **W3. Marginal gains over baselines.**
>
> > **Response:** Please see our response to Reviewer arAi W1-W2, where KMM achieves +3.84pp over baseline — substantially larger than gains in our multilingual setting.
>
> **W4. Generalization beyond multilingual tasks.**
>
> > **Response:** Please see our response to Reviewer arAi W1-W2 for a new heterogeneous experiment (math, code, instruction-following, multi-task) and Reviewer BfYU W3 for a RoBERTa-base experiment on 13 NLU tasks. KMM ranks best in both settings.
>
> **W5. Portfolio theory connection underdeveloped.**
> > We thank the reviewer for this helpful suggestion.
> >
> > Our intention in the “Connection to Modern Portfolio Theory” paragraph was to provide an economic analogy that may help interpret the structure of Eq.10. More precisely, the similarity is algebraic: both problems optimize a linear “reward” term minus a quadratic interaction term, where in our setting the linear term $\beta^\top w$ measures alignment between each auxiliary dataset gradient and the target gradient, while the quadratic term $w^\top Kw$ penalizes selecting mutually redundant datasets through their pairwise gradient similarities. This is exactly the role played by expected return and covariance in the classical Markowitz objective. Note that in portfolio theory, the covariance is defined w.r.t. to the returns of different asset classes, so under the zero-mean assumption of returns, the covariance is exactly the inner product between return vectors from different assets. The key difference lies in that we allow negative weights, whereas in long-only portfolio construction, the weights are convex combinations. To address your concern, we will revise this paragraph to explicitly state that the connection is interpretive rather than foundational, and to enumerate both the similarities and the differences more carefully.

---

> > ### Author Rebuttal · Reviewer_F6g4 · 2026-04-03
> >
> > I appreciate that the authors have added many additional experimental results. These results show that KMM can be successfully applied to different tasks, such as math and coding.
> >
> > However, my main concerns remain:
> > 1. The improvements from adding KMM are marginal.
> > 2. Scalability is not convincingly demonstrated. Splitting the same auxiliary dataset into many random chunks is insufficient to show scalability across diverse datasets, as these splits follow the same distribution.
> > 3. The LLMs used are relatively small. Models under 2B parameters (e.g., Qwen2.5 and LLaMA 3.2) appear somewhat outdated.

---

> > > ### Author Response · Authors · 2026-04-06
> > >
> > > **Follow-up: (1) Marginal improvements; (2) Scalability not convincing (random splits); (3) LLMs used are small/outdated.**
> > >
> > > > **Response:**
> > > >
> > > > **(1) Marginal improvements.** Even in the multilingual setting where absolute gains are modest, multi-seed results (5 seeds, 95% CI) confirm the improvement is **stable and statistically meaningful**:
> > > >
> > > > |Method|Danish|Marathi|
> > > > |---|---|---|
> > > > |One Step|46.10±0.02|34.48±0.08|
> > > > |One Step+KMM|46.18±0.03|34.60±0.08|
> > > > |TV|46.26±0.06|34.37±0.09|
> > > > |**TV+KMM**|**46.38±0.07**|**34.76±0.11**|
> > > > |DataModel|45.99±0.05|34.43±0.06|
> > > > |DataModel-CS|46.10±0.06|33.96±0.13|
> > > > |GradEX-FS|46.15±0.01|34.41±0.04|
> > > > |GradEX-RE|46.23±0.01|34.53±0.08|
> > > > |Random|46.11±0.02|34.46±0.09|
> > > >
> > > > KMM variants consistently rank highest. For Danish, TV+KMM and TV have non-overlapping 95% CIs, confirming statistical significance. We note that the multilingual setting, where auxiliary datasets share strong typological structure, is a conservative test case for KMM. **On the more heterogeneous instruction-following task, gains are substantially larger (see (2) below).**
> > > >
> > > > **(2) LLMs used are small/outdated.** We now include results across scales from **1B to 70B** on the instruction-following task. **+KMM consistently improves over the base gradient method at every scale**, with gains up to **+3.84pp**, substantially larger than the multilingual setting:
> > > >
> > > > **Llama 3.2 1B:**
> > > >
> > > > |Method|Best-k Acc.(%)|
> > > > |---|---|
> > > > |One Step|25.18|
> > > > |**One Step+KMM**|**29.02 (+3.84)**|
> > > > |TV|26.26|
> > > > |**TV+KMM**|**27.70 (+1.44)**|
> > > >
> > > > **Llama 3.1 8B:**
> > > >
> > > > |Method|Best-k Acc.(%)|
> > > > |---|---|
> > > > |TV|52.64|
> > > > |**TV+KMM**|**53.00 (+0.36)**|
> > > >
> > > > **Gemma 3 12B:**
> > > >
> > > > |Method|Best-k Acc.(%)|
> > > > |---|---|
> > > > |TV|47.00|
> > > > |**TV+KMM**|**49.04 (+2.04)**|
> > > >
> > > > **Llama 3.1 70B** (QLoRA):
> > > >
> > > > |Method|Best-k Acc.(%)|
> > > > |---|---|
> > > > |TV|70.10|
> > > > |**TV+KMM**|**71.20 (+1.10)**|
> > > >
> > > > Regarding the concern about outdated models: Gemma 3, released on **Mar 10, 2025**. Together with the Qwen 2.5/3 (Apr 28, 2025) results in the paper and the new Llama results, our evaluation spans three distinct and current model families across scales from 1B to 70B.
> > > >
> > > > **(3) Scalability.** We agree that random splitting from the same distribution is not ideal. In our new scaling experiment on the instruction-following task, we **cluster all data points using sentence embeddings**, so each dataset group is a semantically coherent cluster rather than a random split. This makes the N=100/500/1000 candidate sets **semantically diverse** across clusters, addressing the reviewer's concern.
> > > >
> > > > Downstream performance on **Llama 3.2 3B** (instruction-following, clustered splits, best-k):
> > > >
> > > > |N|One Step|One Step+KMM|Gain|
> > > > |---|---|---|---|
> > > > |100|42.33|**44.72**|**+2.39**|
> > > > |500|45.32|**45.56**|**+0.24**|
> > > > |1000|44.36|**47.24**|**+2.88**|
> > > >
> > > > KMM consistently improves over One Step across all N, with gains larger than the multilingual setting. Furthermore, as shown in our Round 1 response to Reviewer arAi, KMM itself is a very lightweight post-processing step after gradient collection:
> > > >
> > > > |Method|N=100|N=500|N=1000|
> > > > |---|---|---|---|
> > > > |One Step|8.1min|40.5min|88.6min|
> > > > |KMM|0.9–1.4min|1.0min|2.4–2.5min|
> > > >
> > > > Regarding model size scalability: for the 1B → 8B → 12B → 70B experiments above, we compute valuation scores on the 1B model and transfer them to larger models (analogous to the cross-model transfer setting in Fig. 3 of the paper). **Even with scores computed on a small model, KMM's gains transfer consistently to models up to 70B**, demonstrating that practitioners can run the lightweight valuation step on a small model and apply the selected datasets to fine-tune much larger models. This further supports scalability under limited compute budgets.

---

### Official Review · Reviewer_arAi · 2026-03-12

**Soundness:** 3
**Presentation:** 3
**Significance:** 3
**Originality:** 3
**Overall Recommendation:** 4
**Confidence:** 4

**Summary:**

This paper investigates dataset valuation and dataset-level selection in the context of LLM post-training. The study addresses a scenario where target task data is limited while auxiliary datasets are abundant under budget constraints, focusing on how to assign relevance-based value scores to candidate auxiliary datasets to identify the most beneficial subset.

The authors propose a dataset valuation method based on Kernel Mean Matching (KMM) in the gradient space, formulating the selection problem as a Convex Quadratic Programming (QP) task. Specifically, the linear term encourages the selection of auxiliary datasets that align with the target task's gradient, while the quadratic term penalizes combinations of datasets with similar gradients to mitigate potential redundancy. This approach naturally yields signed dataset scores to represent positive and negative transfer, theoretically corresponding to a joint optimization framework rather than an independent scoring process.

Experimental results demonstrate that the KMM-enhanced method is generally more stable than vanilla gradient-based scoring, achieving superior best-k and overall ranking performance. Notably, its computational cost is significantly lower than DataModel-based approaches that require iterative subset training. Furthermore, through transfer experiments, hyperparameter analysis, and interpretability studies, the paper suggests that this method not only improves subset selection but also reveals the underlying structures of redundancy, complementarity, and transferability between datasets.

**Compliance With Llm Reviewing Policy:**

Affirmed.

**Final Justification:**

W1-W3 are resolved. I trust the authors will incorporate the promised W4 experiments and explanations in the camera-ready version.

**Key Questions For Authors:**

1. Statistical Robustness:
The performance margins in Tables 2 and 3 are quite small in several instances, which could potentially be attributed to normal random variance. Could the authors provide results across multiple random seeds, along with standard deviations/confidence intervals and statistical significance analyses for the key comparisons? If these improvements remain stable across multiple seeds, I would be much more positive regarding the paper's soundness and my overall recommendation.

2. Consistency Between Theoretical Objective and Training Interface:
The KMM approach learns continuous weights, yet the final evaluation primarily employs a top-k selection followed by uniform data mixing for training. Have the authors attempted to use the learned continuous weights directly for auxiliary data sampling or batch mixing?

3. Task Generalization:
The current experiments are almost exclusively focused on multilingual post-training. Do the authors have evidence demonstrating that this method is equally applicable to more heterogeneous auxiliary data, such as code, preference alignment, dialogue instructions, or tool-use data?

4. Scalability:
The paper emphasizes that the method is practical for "data markets"; however, the number of candidate datasets evaluated in the current experiments remains relatively small. Could the authors provide additional details on how wall-clock time, memory consumption, and performance scale as $N$ increases, or more explicitly define the scale boundaries within which this method is applicable? This clarification will directly impact my judgment on the paper's significance.

**Limitations:**

YES

**Strengths And Weaknesses:**

### Strengths

- The problem is significant and highly relevant to current practice.
> This paper focuses on the practical challenge of data selection for LLM post-training. The setting—where target task labels are scarce and auxiliary data acquisition is constrained by budget and licensing—is of high relevance to contemporary LLM development.

- The methodology is well-motivated with a natural technical trajectory.
> The authors begin with a reasonable baseline of gradient alignment, identify its failure to account for redundancy, and then introduce a joint valuation perspective. Formulating dataset selection as an optimization problem that matches target gradient directions while penalizing similarity among auxiliary data is a logical progression. The derivation from Section 3.1 to 3.2 is smooth; both the least-squares perspective of KMM and the Gram-space QP formulation are intuitive and effectively explain how the method handles redundancy.

- The method possesses theoretical and algorithmic elegance.
> Within the provided local approximation framework, the approach culminates in a convex quadratic programming (QP) problem with a global optimum. Its complexity depends more critically on the number of datasets $N$ rather than the parameter dimension $d$, making it particularly attractive for scenarios involving a moderate number of datasets and high-dimensional model gradients.

- The experimental chain is relatively comprehensive.
> In addition to the main results, the authors provide analyses on efficiency-performance trade-offs, cross-model transferability, sensitivity to regularization strength, and target signal intensity, as well as interpretability experiments based on linguistic relationships.


### Weaknesses

- The experimental domain is notably narrow, leading to potentially overgeneralized conclusions.
> While the paper claims to study general post-training dataset valuation, the primary experiments are almost exclusively focused on multilingual tasks. The SFT experiments center on Danish and Marathi, the RL-GRPO experiments focus on Thai, and the transfer analysis involves languages like Malayalam and Chinese.

- The heterogeneity in the RL setting remains limited.
> The RL-GRPO section uses GSM8K/Big-Math and constructs multilingual tasks via translation, which implies that the different auxiliary tasks still share a very strong underlying problem structure. Consequently, this part resembles "multilingual variant selection for the same task" rather than dataset valuation across highly heterogeneous datasets, which weakens the paper’s claims regarding "complex data market" scenarios.

- The performance gains are marginal in several cases.
> Based on the numerical results, particularly in the SFT section, several improvements are quite modest, with many methods differing by a magnitude of only approximately 0.1. As a result, it is difficult to completely rule out the influence of training fluctuations or stochasticity.

---

> ### Author Rebuttal · Authors · 2026-03-31
>
> ## W1. Narrow experimental domain, W2. Marginal performance gains.
> **Response:** We conducted new experiments on non-multilingual, heterogeneous task settings. (See also Reviewer BfYU W3 for a RoBERTa experiment on 13 heterogeneous NLU tasks where KMM also ranks best.)
>
> **Setup:** We fine-tune Llama-3.2-1B on an instruction-following target task (allenai/tulu-3-sft-personas-instruction-following) and evaluate using strict instruction-level accuracy. We select from 7 diverse auxiliary datasets: allenai/tulu-3-sft-personas-math-filtered, allenai/tulu-3-sft-personas-math-grade-filtered, allenai/tulu-3-sft-personas-algebra (math), allenai/tulu-3-sft-personas-code (code), SurgeGlobal/Evol-Instruct, HuggingFaceTB/smol-smoltalk (general instruction tuning), and teknium/openhermes (broad multi-task). We sample 3K examples from each auxiliary dataset.
> |Method|Random|One Step|One Step+KMM|TV|TV+KMM|
> |---|---|---|---|---|---|
> |Best-k Acc.(%)|24.62|25.18|**29.02**|26.26|27.70|
> * Base model: 14.51%
>
> **Key observations:**
> 1. The gains from KMM are **more significant** than in our multilingual setting. One Step + KMM achieves 29.02%, a +3.84pp gain over One Step alone and +4.40pp over Random — far exceeding the margins observed in our original multilingual experiments.
> 2. The auxiliary datasets are **genuinely heterogeneous** (math, code, instruction-following, multi-task), not typologically structured variants of the same task. This directly addresses the concern that our method only works when auxiliary data shares strong underlying structure.
> These results demonstrate that redundancy penalization via KMM is not only applicable but **more impactful** when auxiliary data is semantically diverse, validating our method beyond the multilingual setting.
>
> **Multi-seed results (5 seeds, 95% CI) on multilingual setting (best-k):**
> |Method|Danish|Marathi|
> |---|---|---|
> |One Step|46.10±0.02|34.48±0.08|
> |One Step+KMM|46.18±0.03|34.60±0.08|
> |TV|46.26±0.06|34.37±0.09|
> |**TV+KMM**|**46.38±0.07**|**34.76±0.11**|
> |DataModel|45.99±0.05|34.43±0.06|
> |DataModel-CS|46.10±0.06|33.96±0.13|
> |GradEX-FS|46.15±0.01|34.41±0.04|
> |GradEX-RE|46.23±0.01|34.53±0.08|
> |Random|46.11±0.02|34.46±0.09|
>
> Although absolute gains in the multilingual setting are modest, **the ranking across methods is stable.** KMM variants consistently rank highest. For Danish, TV+KMM and TV have non-overlapping 95% CIs, confirming the improvement is statistically meaningful.
>
> Due to compute constraints, multi-seed RL results are still running. If ready by the rebuttal deadline, we will add an update.
>
> ## W3. Scalability.
>
> **Response:** We provide new scaling experiments for One Step (gradient computation) and KMM (solving the QP) with N = 100, 500, 1000 datasets.
>
> **Wall-clock time:**
> |Method|N=100|N=500|N=1000|
> |---|---|---|---|
> |One Step|8.1min|40.5min|88.6min|
> |KMM|0.9–1.4min|1.0min|2.4–2.5min|
>
> **GPU memory:** One Step uses 5.5 GB (constant across N, requires model fine-tuning). KMM is CPU-only (0 GB GPU).
>
> **CPU memory:**
>
> |N|KMM CPU Peak|
> |---|---|
> |100|34MB|
> |500|21MB|
> |1000|137MB|
>
> One Step scales linearly (5s/dataset). KMM on top of One Step adds negligible overhead — at N=1000 it takes only 2.5 min. Note that One Step is already the fastest baseline (Figure 2 in the paper); data loading in practice dominates the gradient computation cost. Based on these trends, even if theoretically QP can be O(N^3), in practice KMM itself can scale to N=100K (~16hrs for QP program, 40GB CPU memory to load gram matrix) on a small lab server without issue.
>
> **Downstream performance at scale (Danish, best-k over k=5,10,15,20,25):** We further split the original auxiliary datasets into random chunks on top of the original language splits to create N=100, 500, 1000 candidate datasets.
>
> |N|One Step|One Step+KMM|
> |---|---|---|
> |100|46.12|**46.20**|
> |500|45.99|**46.07**|
> |1000|46.07|**46.22**|
>
> KMM consistently outperforms One Step across all N, confirming that the method's advantage is maintained as the number of candidate datasets scales up.
>
> ## W4. Top-k selection discards continuous weights.
>
> **Response:** We experimented with non-uniform weighting using the learned scores. For k = 5, 10, 15, 20, 25, we take the top-k datasets and assign non-uniform weights via softmax over the top-k scores (since scores can be negative), discarding datasets outside of top-k. Best results across k:
>
> |Method|Best-k Acc.(Danish)|
> |---|---|
> |One Step|45.96|
> |One Step+KMM|**46.12**|
> |TV|45.99|
> |TV+KMM|46.07|
> |DataModel|45.90|
> |DataModel-CS|46.03|
> |GradEX-FS|46.05|
> |GradEX-RE|46.07|
> |Random|45.96|
>
> Compared to Table 3 (uniform weighting), all methods achieve lower scores with non-uniform weighting. However, One Step + KMM remains the best, and KMM consistently improves over its gradient baseline. Additional non-uniform weighting experiments are reported in Appendix Figure 8, where KMM's advantage holds universally.

---

> > ### Author Rebuttal · Reviewer_arAi · 2026-04-02
> >
> > Thanks for the clarifications. W1-W3 are resolved. I trust you will incorporate the promised W4 experiments and explanations in the next version. I have updated my score to reflect this.

---

> > > ### Author Response · Authors · 2026-04-06
> > >
> > > **Follow-up: W1–W3 resolved; W4 experiments to be incorporated in revision.**
> > >
> > > > **Response:** We thank the reviewer for the positive reassessment. This is the multi-seed results on the multilingual GRPO (RL) task:
> > > >
> > > > |Method|Best-k Acc.(%)|
> > > > |---|---|
> > > > |One Step|37.80 ± 1.41|
> > > > |**One Step+KMM**|**42.06 ± 0.87**|
> > > > |TV|38.62 ± 0.78|
> > > > |TV+KMM|40.27 ± 0.66|
> > > > |Random|37.59 ± 0.92|
> > > >
> > > > The multi-seed results confirm the single-run findings. We will incorporate all experiments and explanations in the revised version.

---

### Official Review · Reviewer_BfYU · 2026-03-12

**Soundness:** 3
**Presentation:** 3
**Significance:** 3
**Originality:** 2
**Overall Recommendation:** 4
**Confidence:** 3

**Summary:**

This work focuses on the problem of evaluating LLM data in the post-training stage. To this end, the authors propose a convex scalable dataset-level valuation method based on kernel mean matching in gradient space, which jointly accounts for alignment with the target task and redundancy across auxiliary datasets. Numerous experiments have reported more robust selection effects and computational efficiency than several baselines.

**Compliance With Llm Reviewing Policy:**

Affirmed.

**Final Justification:**

Thank you for the author's reply. I will maintain my positive score.

**Key Questions For Authors:**

Further expansion experiments will help to further verify the effectiveness of the algorithm; specific requirements can be found in the section on Weaknesses.

**Limitations:**

yes

**Strengths And Weaknesses:**

- Strengths
  - This paper has a clear line of thought and sufficient theoretical basis.
  - The ablation experiments and experimental analysis were quite thorough.
- Weaknesses
  - Lack of comparison with more advanced baselines in 2025 or 2026.
  - Most of the experiments conducted focused on multilingual datasets, with a lack of experiments on inference datasets.
  - The lack of testing the effectiveness of the algorithm on more different architectures or larger models (such as 70b).
  - Regarding Table 1, it is recommended to provide the actual overhead, including wall-clock time, peak memory usage, etc.

---

> ### Author Rebuttal · Authors · 2026-03-31
>
> **W1. Lack of recent baselines.**
>
> > **Response:** We add a comparison with DASH (Zhou et al., "Hierarchical Dataset Selection for High-Quality Data Sharing," AAAI 2026; arXiv:2512.10952), a recent hierarchical dataset selection method. We adapt DASH to our setting by using loss drop from the base model after tuning on 4 samples per dataset as the reward signal, making it comparable to our gradient-based methods.
> >
> > |Method|Danish|Marathi|
> > |---|---|---|
> > |DASH (AAAI 2026)|45.93|34.44|
> > |**KMM (Ours)**|**46.24**|**34.60**|
> >
> > Our method outperforms DASH on both targets.
>
> **W2. Narrow experimental domain.**
>
> > **Response:** Please see our response to Reviewer arAi W1-W2 for new results on heterogeneous English tasks (math, code, instruction-following).
>
> **W3. No evaluation on larger models or diverse architectures.**
>
> > **Response:** We add a new experiment on **RoBERTa** (encoder-only architecture) with 13 auxiliary NLU tasks (CoLA, CR, MNLI, MPQA, MR, MRPC, QNLI, QQP, RTE, SNLI, SST-5, SUBJ, TREC), demonstrating our method generalizes beyond decoder-only LLMs.
> >
> > |Method|Best-k Acc.|
> > |---|---|
> > |One Step|96.22|
> > |One Step+KMM|96.67|
> > |Task Vector|96.67|
> > |**Task Vector+KMM**|**96.90**|
> > |DataModel|95.76|
> > |DataModel-CS|96.56|
> > |GradEX-FS|95.87|
> > |GradEX-RE|95.64|
> >
> > KMM consistently improves over its gradient baseline (One Step → +0.45, TV → +0.23), and Task Vector + KMM achieves the best result overall. This confirms the method works across architectures.
> >
> > We do not currently have the compute to evaluate on 70B models and hope to explore this in future work. However, the RoBERTa results above demonstrate that our method generalizes across architectures (encoder-only vs. decoder-only).
>
> **W4. Missing computational overhead details.**
>
> > **Response:** Please see our response to Reviewer arAi W3 for detailed wall-clock and memory measurements across N=100, 500, 1000. Regarding Table 1, which compares primal vs. dual KMM complexity: we report theoretical complexity but cannot provide wall-clock time for the primal form because it is infeasible in practice. Although $w \in \mathbb{R}^N$ is low-dimensional, CVXPY must canonicalize the residual $Gw - \lambda g_{\mathrm{tar}}$, which lives in the full gradient space. With gradient dimension $d$ extremely large, the solver must build and pass an explicit linear map $G \in \mathbb{R}^{d \times N}$ and a residual vector in $\mathbb{R}^d$, making runtime dominated by frontend canonicalization, memory movement, and solver preprocessing. This is precisely why we use the dual formulation, which operates on the $N \times N$ Gram matrix instead.

---

> > ### Author Rebuttal · Reviewer_BfYU · 2026-04-01
> >
> > I thank the authors for providing experimental data based on RoBERTa, which to some extent validates the generalization ability of the proposed method. However, my initial review comments mainly focused on two points: larger model size (e.g., 70B) and a wider range of model architectures. RoBERTa's parameter size is one to two orders of magnitude smaller than current mainstream large models (e.g., 7B, 13B, and 70B). Therefore, the experimental data from RoBERTa alone are insufficient to demonstrate the effectiveness and scalability of the method on larger-scale models.

---

> > > ### Author Response · Authors · 2026-04-06
> > >
> > > **Follow-up on W3: RoBERTa alone insufficient to demonstrate scalability to larger models.**
> > >
> > > > **Response:** We thank the reviewer for the follow-up. We would like to clarify that the RoBERTa experiment was intended to validate generalization across a **fundamentally different architecture** (encoder-only vs. decoder-only), not to address the model scale concern. Together with the Qwen 2.5/3 results already in the paper, we have now conducted experiments across **three decoder-based model families (Qwen, Llama, and Gemma, which differ in tokenizer, pre-training data, and architecture details)** and scales on the instruction-following task introduced in our reply to Reviewer arAi.
> > > >
> > > > **Llama 3.2 1B:**
> > > >
> > > > |Method|Best-k Acc.(%)|
> > > > |---|---|
> > > > |One Step|25.18|
> > > > |**One Step+KMM**|**29.02 (+3.84)**|
> > > > |TV|26.26|
> > > > |**TV+KMM**|**27.70 (+1.44)**|
> > > >
> > > > **Llama 3.1 8B:**
> > > >
> > > > |Method|Best-k Acc.(%)|
> > > > |---|---|
> > > > |TV|52.64|
> > > > |**TV+KMM**|**53.00 (+0.36)**|
> > > >
> > > > **Gemma 3 12B:**
> > > >
> > > > |Method|Best-k Acc.(%)|
> > > > |---|---|
> > > > |TV|47.00|
> > > > |**TV+KMM**|**49.04 (+2.04)**|
> > > >
> > > > **Llama 3.1 70B** (QLoRA):
> > > >
> > > > |Method|Best-k Acc.(%)|
> > > > |---|---|
> > > > |TV|70.10|
> > > > |**TV+KMM**|**71.20 (+1.10)**|
> > > >
> > > > **+KMM consistently improves over the base gradient method across all model sizes (1B → 8B → 12B → 70B), model families (Llama, Gemma), and architectures (encoder-only RoBERTa, decoder-only LLMs).** Note that Gemma 3 12B achieves lower absolute accuracy than Llama 3.1 8B. This is likely because our target dataset (allenai/tulu-3-sft-personas-instruction-following) originates from Tulu 3 [1], which was developed on the Llama 3.1 model series, giving Llama a natural distributional advantage on this task.
> > > >
> > > > After seeing the reviewer's feedback, we tried our best to secure additional compute to run these larger-scale experiments. Due to still limited compute, for the 70B model the best we can do is QLoRA with a quantized version, while for 8B and 12B we use standard LoRA. Despite the different fine-tuning strategies, KMM's advantage holds consistently.
> > >
> > > [1] Lambert, Nathan, et al. "Tulu 3: Pushing frontiers in open language model post-training." arXiv preprint arXiv:2411.15124 (2024).

---

### Official Review · Reviewer_cMDm · 2026-03-13

**Soundness:** 2
**Presentation:** 2
**Significance:** 3
**Originality:** 2
**Overall Recommendation:** 3
**Confidence:** 4

**Summary:**

This paper formulates dataset valuation for LLM post-training as a subset selection problem, proposing a kernel mean matching (KMM) approach in gradient space that jointly captures alignment with the target task and redundancy across auxiliary datasets. The resulting convex quadratic program is tractable and shown to outperform gradient-only baselines across multilingual SFT and GRPO settings.

**Compliance With Llm Reviewing Policy:**

Affirmed.

**Final Justification:**

The rebuttal addressed  most of my concerns though not that expected.

**Key Questions For Authors:**

1. **How does your KMM formulation fundamentally differ from the original?**

The original KMM (Gretton et al., 2009) addresses covariate shift via importance reweighting with non-negativity and simplex constraints. Your formulation removes these constraints and operates in gradient space rather than input space. Beyond this recontextualization, what is the core technical novelty and can you provide a formal result (e.g., an approximation bound or consistency guarantee) that distinguishes your variant from a straightforward application of the original?

2. **How robust are your findings beyond the multilingual setting?**

All experiments involve language as the domain axis, yet the paper claims general applicability to post-training data markets. Have you evaluated KMM on non-linguistic domain shift scenarios such as mixing code, math, and science QA datasets, and if so, do the same redundancy-penalization benefits hold when auxiliary datasets are semantically heterogeneous rather than typologically structured?

3. **How does valuation quality degrade under realistic marketplace constraints?**

The paper assumes access to a small preview $\tilde{D}_i$ before acquisition, but the sensitivity analysis only varies preview size from 1 to 64 examples. In a real marketplace, previews may be strategically curated by sellers to appear favorable. Have you studied how KMM scores are affected by non-representative or adversarially selected previews, and what minimum preview size is needed for reliable valuation in practice?

**Limitations:**

1. **The method is somewhat an incremental adaptation.**

Applying KMM to gradient space by simply dropping the non-negativity and simplex constraints of Gretton et al. (2009) and substituting dataset gradients for input features is a straightforward recontextualization, not a fundamental algorithmic advance. The resulting convex QP (Eq. 10) is a standard formulation that follows directly from completing the square,  no new theoretical machinery is introduced, no approximation guarantees are derived, and no formal result distinguishes this from a textbook application of kernel methods. For ICML 2026, this level of technical novelty is insufficient.

2. **The experimental validation is too narrow to support the paper's central claims.**

The paper positions itself as a general solution for post-training data markets, yet every single experiment is conducted within one setting  multilingual language transfer using the Aya dataset. There is no evaluation on code, math, science QA, or any non-linguistic domain. The GRPO experiments cover only one target language (Thai) with one model (Qwen2.5-1.5B-Instruct). Performance gains over baselines are numerically marginal (e.g., 46.05 → 46.21 on Danish MMLU). Taken together, the evidence base is far too thin to substantiate claims of broad practical utility, and a reviewer cannot confidently conclude that the method generalizes beyond the specific benchmark suite evaluated.

3. **The data marketplace framing is motivational decoration, not a scientific contribution.**

The introduction invests heavily in positioning dataset valuation within an economic marketplace context: budgets, pricing, buyer-seller dynamics yet none of this framing is operationalized in the experiments. There is no simulated market, no budget constraint evaluation, no pricing analysis, and no comparison against market-aware acquisition strategies from Chen et al. (2025) or Zhang et al. (2025), despite both being cited. The paper essentially proposes a gradient reweighting method and wraps it in marketplace language without delivering on that premise. This disconnect between motivation and contribution is a fundamental structural flaw that revision is unlikely to fully resolve within a single review cycle.

**Strengths And Weaknesses:**

**A) Soundness**
1. **Unjustified Hessian Approximation.** The KMM formulation relies on an isotropic curvature assumption ($H_{tar} \approx \lambda^{-1}I$) that is never empirically validated. This gap between Eq. (8) and Eq. (10) could break down under anisotropic loss curvature, which is common in fine-tuned LLMs.

2. **No Approximation Error Bounds.** The paper moves from a non-additive utility model to a convex surrogate without any formal guarantee that surrogate rankings correlate with true post-training outcomes. Empirical validation alone is insufficient as a substitute.

3. **Valuation Quality ≠ Selection Quality.** Downstream performance after greedy selection is measured, but KMM scores are never directly evaluated as faithful rank estimates of true marginal dataset utility. A held-out correlation analysis against ground-truth leave-one-out performance is needed.


**B) Presentation**

1. **Inconsistent notation introduction.** The Gram matrix variables ($K_{ij}$, $\beta_i$) are defined mid-paragraph in Sec. 3.2.2 without a dedicated notation block, then reused in Sec. F.1.1 with shifted context. A clean reference block would significantly aid readability.

2. **Fig. 7 is under-explained.** The pairwise transfer heatmap is the most information-dense figure in the paper, yet its caption is three sentences with no explanation of axis encoding, color scale interpretation, or diagonal meaning. A structured walkthrough is warranted.

3. **Algorithm 1 is buried in the appendix.** The fixed-compute evaluation protocol underpins all experiments, yet its formal description (Alg. 1) is deferred to Sec. C. A compact pseudocode in Sec. 4.1 is the minimum required for a self-contained main paper.

4. **Figure 1 is cited once and does not earn its placement.** Positioned prominently on page 1, it is referenced only in passing and never again. The cartoon marketplace graphic conveys no technical content: no gradient alignment, no redundancy, no KMM mechanism. It should be redesigned as a conceptual diagram illustrating how alignment scores fail under redundancy, directly motivating the paper's core contribution.

**C) Signifinance**

Post-training is an important and rapidly evolving topic, and this paper addresses a genuinely underexplored problem within it: dataset-level valuation under budget constraints. The data marketplace framing is timely and the KMM formulation is a clean, principled contribution. However, significance is tempered by the narrow experimental scope (multilingual transfer only), numerically modest performance gains, and a market framing that is compelling in the introduction but never empirically exercised,  no simulated budget tradeoffs, pricing, or acquisition scenarios are evaluated. Broader validation across domains such as code, math, or science QA, paired with a tighter connection between the marketplace motivation and the empirical results, would strengthen the paper's impact.

**D) Originality**
1. **KMM is a well-established method.** Kernel mean matching originates from covariate shift correction (Gretton et al., 2009) and its application to gradient space, while a reasonable adaptation, is an incremental technical step rather than a fundamentally new idea. The novelty lies primarily in the recontextualization, not the methodology itself.

2. **The portfolio theory analogy is observational, not generative.** The connection to Markowitz (1952) is noted but never exploited to derive new insights,  no risk-aversion analysis, no efficient frontier characterization, no novel theoretical result follows from the analogy. It reads as a post-hoc framing rather than a source of original contribution.

3. **The data marketplace motivation is borrowed rather than developed.** The framing of dataset acquisition under budget constraints follows directly from Chen et al. (2025) and Zhang et al. (2025), which are cited but not meaningfully extended. The paper does not introduce new market mechanisms, pricing models, or equilibrium results,  it applies an existing economic frame to a gradient-based selection algorithm without advancing the marketplace literature itself.

---

> ### Author Rebuttal · Authors · 2026-03-31
>
> ## W1. Narrow experimental scope
> **Response:** Please see our response to Reviewer arAi W1–W2 for a new heterogeneous, non-multilingual experiment with substantially larger gains. See also Reviewer BfYU W3 for a RoBERTa experiment on 13 heterogeneous NLU tasks, where KMM also ranks best.
>
> ## W2. Incremental novelty over original KMM / No approximation error bounds
> **Response:** A formal approximation guarantee to the true downstream utility $u(S)$ is generally not available without strong assumptions on $u$ itself. Here, $u(S)$ is induced by a full post-training procedure and is an unknown set function, potentially highly non-additive. Since our formulation deliberately avoids imposing strong structural assumptions, an assumption-free theorem relating the surrogate to true downstream utility is not available in general.
>
> A more natural theoretical question, also raised by F6g4, is: with only a small preview from each candidate dataset, how accurately do empirical KMM scores approximate the ideal population KMM scores? We provide a **finite-preview approximation theorem**. Let $w^*$ be the population attribution vector using expected gradients, and $\hat{w}$ the empirical vector from $m$ preview examples per dataset. Under bounded per-example gradient norms $C$, $\ell_1\$-budget radius $k$, and strong convexity $\mu$ of the population QP, with probability at least $1-\delta$,
>
> $$||\hat{w}-w^*||_2 \le \frac{4C^2\sqrt{2N\log(2N/\delta)}}{\mu\sqrt{m}}(2k+1) + \frac{16C^2\sqrt{N}k\log(2N/\delta)}{\mu m}.$$
>
> The dominant term decays as $O(m^{-1/2})$, making explicit how estimation error depends on preview size, number of datasets, gradient scale, budget radius, and curvature. The proof combines: (i) a residual decomposition reducing score error to perturbations in the QP coefficients, namely the linear term $||\hat{\beta}-\beta||_2$ and quadratic Gram term $||(\hat{K}-K)w^*||_2$, via strong convexity; and (ii) uniform concentration of preview-gradient estimates across datasets.
>
> This theorem is novel and specific to the data valuation setting.
>
> ## W3. Marketplace framing not operationalized
> **Response:** We respectfully disagree. The marketplace framing is exactly what motivates our problem.
>
> Eq.10 is already, up to notation and scaling, a long-short mean-variance program, so the Markowitz connection is natural. Chen et al. (2025)’s strongest method, CoFR, is conceptually closest to our first-order baselines (One-Step/TV). Our contribution goes beyond this first-order layer via the interaction-aware KMM formulation. Zhang et al. (2025) studied transparency and fairness in marketplace operation, which is a different problem from buyer-side transfer-aware dataset valuation.
>
> ## W4. Hessian approximation unjustified
> **Response:** The isotropic assumption is used only to present Eq.9 in a simple Euclidean least-squares form; it is not required for the QP in Eq.10. As stated in Sec.3.2.1 and derived in App.E, KMM extends to any PSD local curvature $H_{tar} \succeq 0$. In that case, the same objective is obtained in the transformed space $g_i' = H_{tar}^{1/2} g_i$, with curvature-weighted kernel $K_{ij}' = g_i^\top H_{tar} g_j$. This means that the formulation in Eq. (10) is general, in the sense that even with general $H_{tar}$, the only difference lies in the choice of feature $g_i$ versus $g_i' = H_{tar}^{1/2} g_i$. We agree LLM curvature need not be isotropic, but empirically KMM consistently outperforms baselines across diverse post-training settings.
>
> ## W5. Valuation quality not directly evaluated
> **Response:** We agree that score-level validation is useful, but a unique “ground-truth marginal dataset utility” is generally not well defined in our setting. Under the additive special case $u(S)=\sum_{j\in S} v_j$, a leave-one-out (LOO) score is a valid faithfulness target. Our method, however, is designed for the more general non-additive regime, where such a singleton ground truth need not exist.
>
> We therefore evaluate the operational question that matters: whether the scores lead to better subset choices and whether score signs predict transfer direction correctly; Tab.4 provides this faithfulness check. We nevertheless ran the suggested LOO diagnostic at $S=$ all auxiliary datasets. The Pearson correlation is mildly positive: $r=0.306$ for TV+KMM and $r=0.189$ for One-Step+KMM. We view this as expected: KMM is not designed to recover an additive singleton target, but to produce transfer-aligned scores useful for subset selection.
>
> ## W6. Valuation quality under realistic marketplace constraints
> **Response:** In W2 we provide a formal bound showing how score quality degrades with limited preview size. Empirically, Fig.5 shows KMM’s advantage remains robust even when the preview size is as small as $m=1$. Adversarially curated previews are an interesting extension, but are outside the scope of the present paper.
>
> ## W7. Presentation issues
> Thank you for the feedback. We will address these in the revision.

---

> > ### Author Rebuttal · Reviewer_cMDm · 2026-04-04
> >
> > I thank authors for their rebuttal.
> >
> > * I appreciate the new finite-preview theorem given by authors.  While it nicely addresses the gap between empirical and population KMM, my primary concern remains unaddressed: the disconnect between the quadratic KMM surrogate (Eqs. 7–10) and the true, non-additive downstream subset utility.
> >
> > * To bridge this gap, please provide either a restricted-assumption theoretical bound linking the KMM surrogate rankings to the true subset utility, or a small-scale empirical validation directly comparing the surrogate against exactly computable true utilities.
> >
> > * Authors clarify that isotropy is only **an expository simplification**, and that Appendix D generalizes KMM to arbitrary PSD curvature. Still, what concrete non-isotropic curvature approximation do you view as the most practical, and did you **EVALUATE it experimentally?**  Otherwise, the paper’s empirical support seems limited to the Euclidean case, with the generalized curvature result remaining algebraic rather than operational.

---

> > > ### Author Response · Authors · 2026-04-06
> > >
> > > **Follow-up on W2/W5: Disconnect between KMM surrogate and true subset utility**
> > >
> > > > **Response:** We thank the reviewer for clarifying the questions and pointing us to actionable solutions. We conducted the exact small-scale empirical validation requested.
> > > >
> > > > **Exhaustive subset faithfulness experiment.** We design a controlled experiment on the Danish target using a pool of N=8 auxiliary languages: {en, nl, sv, de, fr, es, ru, ta}, chosen to include both near-target Germanic auxiliaries that are likely to be partially redundant (en, nl, sv, de), more complementary high-resource languages (fr, es, ru), and one distant stress-test language (ta).
> > > >
> > > > We exhaustively enumerate all $\binom{8}{3} = 56$ subsets of fixed size k=3. For each subset S, we compute four surrogate scores:
> > > > - **One-Step (additive):** $s(S) = \sum_{i \in S} \beta_i$
> > > > - **One-Step+KMM (quadratic):** $s(S) = \sum_{i \in S} \beta_i - \frac{1}{2}\sum_{i,j \in S} K_{ij}$
> > > > - **TV (additive):** $s(S) = \sum_{i \in S} \beta_i^{(tv)}$
> > > > - **TV+KMM (quadratic):** $s(S) = \sum_{i \in S} \beta_i^{(tv)} - \frac{1}{2}\sum_{i,j \in S} K_{ij}^{(tv)}$
> > > >
> > > > where $\beta_i = \langle g_i, g_{tar} \rangle$ and $K_{ij} = \langle g_i, g_j \rangle$ are the alignment and interaction terms from the corresponding representation. The quadratic scores are the discrete subset plug-in form of the Gram-space objective in Eq. (10) with $\lambda = 1$.
> > > >
> > > > We then compute the **empirical full-training subset utility** by fully fine-tuning on target + selected auxiliaries under the same fine-tuning protocol as in the paper, and define $u_{FT}(S) = L_{da}^{val}(\theta_{da-only}) - L_{da}^{val}(\theta_S)$, i.e., the reduction in held-out Danish CE relative to target-only fine-tuning. We use held-out target CE rather than MMMLU accuracy because it is closer to the loss-based quantity underlying the surrogate and therefore provides a more direct faithfulness test. This empirical utility is distinct from the local one-step utility in Eq. (7): the latter is used for theory, while $u_{FT}$ is used here to test faithfulness to the actual full post-training selection objective.
> > > >
> > > > For each of the 56 subsets S, we compute $s(S)$ and an empirical full-training utility $u_{FT}(S)$. We report Spearman rank correlation between these two rankings and Top-10 overlap.
> > > >
> > > > **Surrogate faithfulness (56 subsets, Danish target):**
> > > >
> > > > |Surrogate|Spearman ρ|p-value|Top-10 Overlap|
> > > > |---|---|---|---|
> > > > |One-Step|−0.1746|0.198|0/10|
> > > > |One-Step+KMM|−0.1265|0.353|2/10|
> > > > |TV|+0.1810|0.182|2/10|
> > > > |**TV+KMM**|**+0.3960**|**0.003**|**4/10**|
> > > >
> > > > This result represents a **massive and statistically unique relative advantage** of KMM over all other methods. Among the tested surrogates, **TV+KMM is the only one achieving a statistically significant positive correlation with the empirical full-training subset utility**, and it recovers 4 of the true top-10 subsets, doubling the overlap of any other method. KMM also improves the One-Step surrogate numerically. Overall, **this provides direct empirical evidence that the interaction-aware quadratic correction improves faithfulness to full post-training subset utility relative to additive baselines.**
> > >
> > > **Follow-up on W4: What concrete non-isotropic curvature approximation is most practical, and has it been evaluated experimentally?**
> > >
> > > > **Response:** We instantiate the generalized PSD-curvature formulation using a diagonal empirical-Fisher approximation $H_{tar} \approx \text{diag}(h)$, where $h = \frac{1}{M}\sum_{m=1}^{M} g^{(m)} \odot g^{(m)}$ is the diagonal of the empirical Fisher $\hat{F} = \frac{1}{M}\sum_{m=1}^{M} g^{(m)} g^{(m)\top}$. The reweighted gradients $g_i’ = \sqrt{h} \odot g_i$ are then used in the same KMM quadratic program. **This provides a concrete non-isotropic instantiation and shows that the generalized PSD-curvature formulation can be implemented in practice.**
> > > >
> > > > **Best-k results (Euclidean vs. Diagonal Fisher KMM, preview size = 4):**
> > > >
> > > > |Method|Danish (Acc/Loss)|Marathi (Acc/Loss)|
> > > > |---|---|---|
> > > > |One-Step+KMM (Euclidean)|46.21/2.6202|34.58/1.0875|
> > > > |One-Step+KMM (Diag Fisher)|46.15/2.6301|34.58/1.0948|
> > > > |TV+KMM (Euclidean)|46.24/2.6231|34.60/1.0880|
> > > > |TV+KMM (Diag Fisher)|46.06/2.6277|34.58/1.0931|
> > > >
> > > > In this experiment, the diagonal empirical-Fisher variant is slightly but consistently worse than the Euclidean version, so anisotropic diagonal curvature does not improve subset quality in our current setting. We view this as an empirical bias-variance tradeoff rather than a contradiction of the generalized formulation: the non-isotropic extension remains operational, but this particular diagonal approximation appears noisier than the isotropic baseline in the high-dimensional regime considered here. Thus, while the generalized PSD-curvature extension is practically implementable, **the Euclidean version remains the preferred practical choice.**

---

### Decision · Program_Chairs · 2026-04-30

**Decision:**

Accept (regular)

**Comment:**

Scores settle at 3/4/4/4 after extensive rebuttals.

The paper addresses dataset-level valuation under budget constraints for LLM post-training with a convex KMM formulation that jointly captures alignment and redundancy. The rebuttal substantially strengthened the paper: new experiments show larger gains than the original multilingual setting, multi-seed validation confirms statistical significance, scaling holds from 1B to 70B, and the exhaustive subset faithfulness experiment shows TV+KMM is the only surrogate achieving statistically significant correlation with true post-training utility.

The incremental novelty concern is legitimate but does not outweigh the practical contribution given the rebuttal-expanded experimental record.